# Cancer immunotherapy by NC410, a LAIR-2 Fc protein blocking human LAIR-collagen interaction

M Ines Pascoal Ramos[1,2†], Linjie Tian[3†], Emma J de Ruiter[4], Chang Song[3], Ana Paucarmayta[3], Akashdip Singh[1,2], Eline Elshof[1,2], Saskia V Vijver[1,2], Jahangheer Shaik[3], Jason Bosiacki[3], Zachary Cusumano[3], Christina Jensen[4], Nicholas Willumsen[4], Morten A Karsdal[4], Linda Liu[3], Sol Langermann[3], Stefan Willems[5§], Dallas Flies[3‡*], Linde Meyaard[1,2‡*]

[1]Center for Translational Immunology, University Medical Center Utrecht, Utrecht University, Utrecht, Netherlands; [2]Oncode Institute, Utrecht, Netherlands; [3]NextCure, Beltsville, United States; [4]Nordic Bioscience, Herlev, Denmark; [5]Department of Pathology, University Medical Center Utrecht, Utrecht University, Utrecht, Netherlands

*For correspondence:
fliesd@nextcure.com (DF);
L.Meyaard@umcutrecht.nl (LM)

[†]These authors contributed equally to this work
[‡]These authors also contributed equally to this work

Present address: [§]Department of Pathology and Medical Biology, University Medical Center Groningen, Groningen, The Netherlands

**Abstract** Collagens are a primary component of the extracellular matrix and are functional ligands for the inhibitory immune receptor leukocyte-associated immunoglobulin-like receptor (LAIR)-1. LAIR-2 is a secreted protein that can act as a decoy receptor by binding collagen with higher affinity than LAIR-1. We propose that collagens promote immune evasion by interacting with LAIR-1 expressed on immune cells, and that LAIR-2 releases LAIR-1-mediated immune suppression. Analysis of public human datasets shows that collagens, LAIR-1 and LAIR-2 have unique and overlapping associations with survival in certain tumors. We designed a dimeric LAIR-2 with a functional IgG1 Fc tail, NC410, and showed that NC410 increases human T cell expansion and effector function in vivo in a mouse xenogeneic-graft versus-host disease model. In humanized mouse tumor models, NC410 reduces tumor growth that is dependent on T cells. Immunohistochemical analysis of human tumors shows that NC410 binds to collagen-rich areas where LAIR-1[+] immune cells are localized. Our findings show that NC410 might be a novel strategy for cancer immunotherapy for immune-excluded tumors.

## Introduction

The introduction of immune checkpoint blockade therapies in the clinic has increased cancer treatment options for a wide range of tumors, leading to unprecedented and long-lasting clinical responses. However, not all patients show the same degree of response and not all tumors respond to these therapies (*Gibney et al., 2016*; *Darvin et al., 2018*). Thus, identifying novel checkpoints and developing ideal combinations of immunotherapies is essential to optimize and enhance the efficacy of treatment for these tumors, achieving durable anti-tumor effects with reduced side effects.

The extracellular matrix (ECM) is a major structural component in all tissues. It comprises a non-cellular meshwork of proteins, glycoproteins, proteoglycans and polysaccharides with collagens as the most abundant proteins. Ongoing ECM remodeling ensures tissue integrity and function, with specific collagens being synthesized and degraded in a highly regulated manner (*Bonnans et al., 2014*). At least 28 different collagens comprising at least 43 genes have been identified (*Myllyharju and Kivirikko, 2004*). The ECM functions not only as a scaffold for tissue organization but also provides critical biochemical and biomechanical cues that instruct cell growth, survival, differentiation and migration and regulate vascular development and immune function (*Hynes, 2009*).

Several epithelial tumors including breast, pancreatic, colorectal, ovarian and lung cancer are characterized by a dense ECM where high collagen content correlates with poor prognosis (*Fang et al., 2014*). Indeed, ECM or 'Matrisome' signatures associated with tumor type and stage of disease have been described (*Levental et al., 2009*; *Lu et al., 2012*; *Naba et al., 2012*; *Naba et al., 2014a*). Cancer-associated fibroblasts (CAFs) (*Kalluri, 2016*), macrophages and tumor cells themselves (*Naba et al., 2012*; *Januchowski et al., 2014*; *Naba et al., 2014a*) all contribute to increased collagen production and remodeling during cancer progression.

Most tumors overexpress a diverse set of collagens in an abnormal fashion (*Nissen et al., 2019*; *Xu et al., 2019*). Overexpression of collagens has been associated with poor overall survival in several tumor types, such as lung (*Iizasa et al., 2004*; *Ke et al., 2014*), colorectal (*Naba et al., 2014b*; *Huang et al., 2018*) and ovarian cancer (*Cheon et al., 2014*; *Wu et al., 2014*). The capacity of tumors to induce remodeling of collagens in the tumor microenvironment (TME) was primarily thought to create a suitable microenvironment for tumor cell growth. We now consider abnormal collagen production, composition and organization in the TME as a cause of immune dysfunction, conveying a chronic-active wound-healing response instead of anti-tumor immune responses necessary for immune surveillance and the eradication of the tumor (*Antonio et al., 2015*; *Rosowski and Huttenlocher, 2015*). The abnormal ECM also builds physical fibrotic barriers to exclude immune cells and therapeutic agents from access to tumor cells (*Henke et al., 2019*).

TME-expressed collagen can interact directly with the inhibitory collagen receptor leukocyte-associated immunoglobulin-like receptor 1 (LAIR-1) (*Rygiel et al., 2011*). LAIR-1 is an immune checkpoint broadly expressed on the cell surface of immune cells (*Meyaard et al., 1997*) that binds to collagen (*Lebbink et al., 2006*) and molecules with collagen-like domains (*Son et al., 2012*; *Olde Nordkamp et al., 2014b*). LAIR-1[+] cells strongly adhere to collagen (*Lebbink et al., 2006*). Upon triggering, LAIR-1 inhibits NK cell (*Meyaard et al., 1997*), T cell (*Maasho et al., 2005*; *Zhang et al., 2014*; *Park et al., 2020*), B cell (*van der Vuurst de Vries et al., 1999*; *Merlo et al., 2005*), monocyte (*Son and Diamond, 2015*) and DC function (*Poggi et al., 1998*; *Son et al., 2012*). Thus, besides formation of an ECM-rich and fibrotic tumor niche, TME-expressed collagens can function to promote immune evasion through their direct interaction with LAIR-1.

LAIR-2 is a natural soluble homolog of LAIR-1 that is present in humans and non-human primates, but not other animals. LAIR-2 shares 83% identity with the LAIR-1 extracellular region, binds to collagens with higher affinity than LAIR-1 and functions as a natural antagonist to block cell membrane LAIR-1 inhibitory signaling (Lebbink, *Lebbink et al., 2008*; *Olde Nordkamp et al., 2011*). We developed a dimeric LAIR-2 Fc fusion protein, NC410, as a novel immunomedicine to both target tumor ECM and promote T cell function through blockade of LAIR-1-mediated inhibition. Hereby, we sought to restore anti-tumor immunity.

## Results

### The collagen:LAIR-1 axis is an immune regulatory pathway to target for cancer immunotherapy

First, we performed a TCGA meta-analysis combining all 28 collagens as a group across 21 TCGA tumor subtypes. When we examined the mRNA expression of all 43 collagen genes combined, 10 out of 21 tumors had reduced overall survival with high (quantile) overall levels of collagens (*Figure 1A*, *Source code 1*). Likewise, the LAIR-1 expression was significantly upregulated in 11 of 21 tumors when evaluated by TCGA data (*Figure 1—figure supplement 1*, *Source code 2* and *3*). LAIR-1 expression is observed in all tumor types and largely restricted to immune cells. By stratifying patients for high- and low-quantile LAIR-1 mRNA expression, we observed that patients with high LAIR-1 mRNA expression had lower survival probability in four tumor types (*Figure 1B*, *Source code 4*). We further stratified the patients in the TCGA database to include high or low expression of all collagens combined with high or low LAIR-1 for overall survival analysis (*Figure 1C*, *Source code 5*). A unique set of tumor types emerged that overlapped with those with significant increased LAIR-1 expression, but only partially with the tumor types in that of the total collagen overall survival analysis. These analyses highlighted the complexity of the TME and also supported targeting the collagen:LAIR-1 axis for cancer immunotherapy.

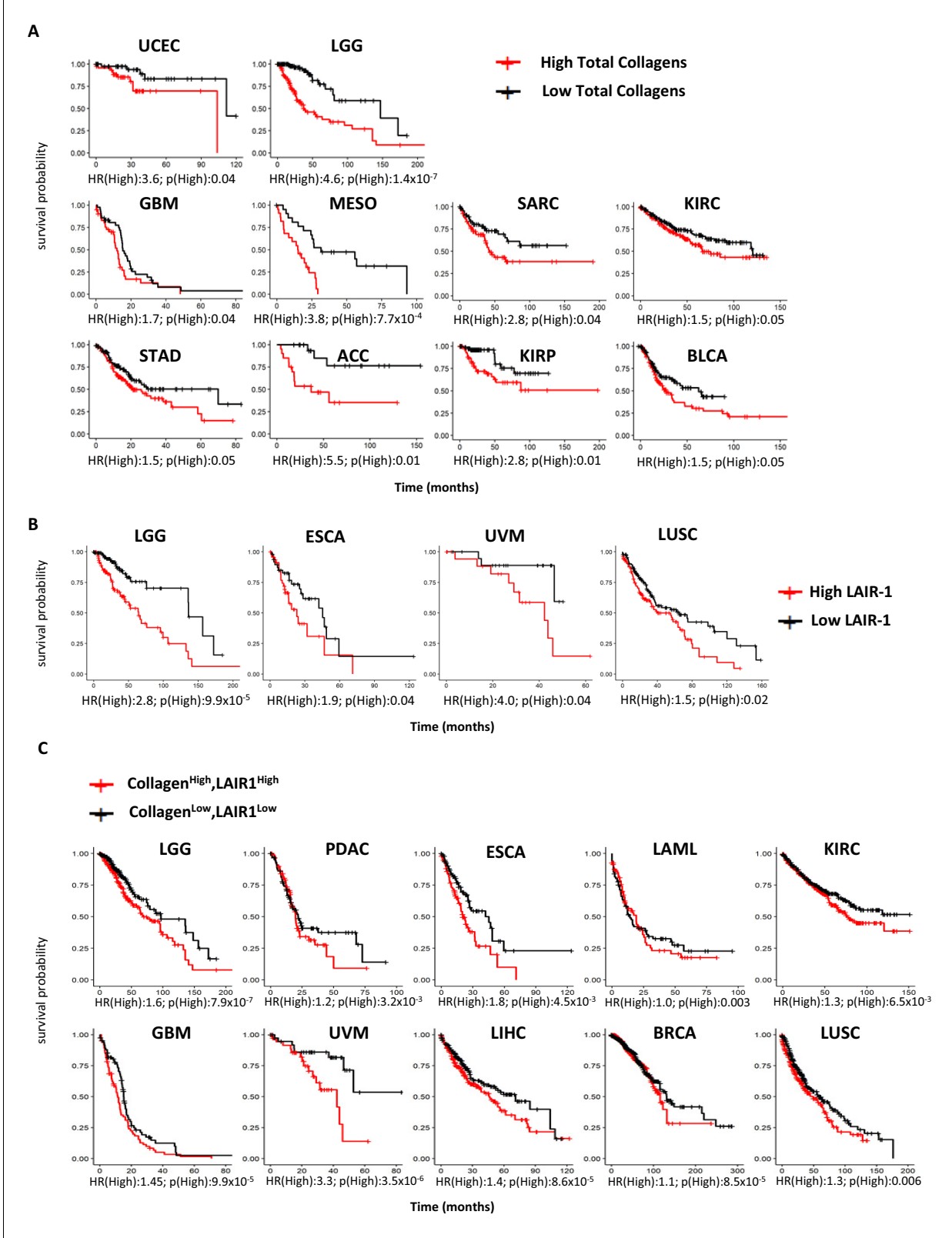

**Figure 1.** High expression of total collagens and LAIR-1 is associated with reduced overall survival. The expression of 43 collagen genes, LAIR-1 and LAIR-2 in normal (gray) and tumor tissue (red) were queried using TCGA database. (**A**) The expression of 43 different collagen genes was assessed together for association with overall survival. The log2 transformed average collagen expression was divided into four quantiles individually for each tumor type. The patients in lower quantile (black) were considered as individuals with low expression and those in the upper quantile (red) were

*Figure 1 continued on next page*

*Figure 1 continued*

considered as those with high expression, respectively. Tumor types in which poor overall survival associated with collagen overexpression are shown: uterine corpus endometrial carcinoma (UCEC), brain lower grade glioma (LGG), glioblastoma multiforme (GBM), mesothelioma (MESO), sarcoma (SARC), kidney renal clear cell carcinoma (KIRC), stomach adenocarcinoma (STAD), adrenocortical carcinoma (ACC), kidney renal papillary cell carcinoma (KIRP) and bladder urothelial carcinoma (BLCA). (B) Patients were grouped into four quantiles with low 25% quantile (black) and high 25% quantile (red) of LAIR-1 mRNA expression compared for overall survival analysis. Esophageal carcinoma (ESCA), uveal melanoma (UVM) and lung squamous cell carcinoma (LUSC). (C) Collagen expression and LAIR-1 expression were assessed together for association with overall survival. Patients in the lower 25% quantile for both average collagen and LAIR-1 were grouped in the collagen[low]LAIR-1[low] (black), and as collagen[high]LAIR-1[high] if they were in the high 25% quantile (red) for both collagen and LAIR-1. Pancreatic adenocarcinoma (PDAC), acute myeloid leukemia (LAML), liver hepatocellular carcinoma (LIHC) and breast invasive carcinoma (BRCA). Hazards ratio indicating if high expression is associated with poor survival (HR (High)), and p-value (p(High)) indicating the significance of association was determined using Wald test as indicated on the x-axis.

The online version of this article includes the following figure supplement(s) for figure 1:

**Figure supplement 1.** Expression of total collagens, LAIR-1 and LAIR-2 in healthy and tumor tissues.

## Development of NC410, a dimeric LAIR-2 Fc fusion protein that blocks collagen interaction with LAIR-1

Soluble LAIR-2 can block immune cell expressed LAIR-1 binding to its ligands (*Lebbink et al., 2008*). We posited that LAIR-2 may correlate with increased overall survival in some tumors. TCGA evaluation demonstrated that six tumor types had improved overall survival with high (quantile) expression of LAIR-2 (*Figure 2A*). This was despite relatively low expression levels of LAIR-2 in most tumors in comparison to LAIR-1 (*Figure 1—figure supplement 1*, *Source code 3*). These data supported supplemental LAIR-2 as a therapeutic intervention that might inhibit LAIR-1 inhibitory interaction with collagen in the TME of solid tumors. Taking advantage of LAIR-2 as a natural decoy system that would both target tumors and reverse immune inhibition, we generated a LAIR-2 human IgG1 Fc fusion protein for therapeutic use named NC410 (*Figure 2B*). This fusion protein exists as a dimeric protein due to the cysteine bonding in the Fc portion of the protein. NC410 bound to human collagen I and III with high avidity (*Figure 2C*). While LAIR-2 is not present in rodents, it is cross-reactive to rat and mouse collagens due to the highly conserved nature of collagens across species (*Figure 2C*). NC410 completely blocked human LAIR-1 binding to collagen I since LAIR-2 binds with higher affinity to collagens than human LAIR-1, as reported previously (*Lebbink et al., 2008*; *Olde Nordkamp et al., 2011*), supporting its potential role as a LAIR-1 antagonist therapeutic (*Figure 2D*). To determine if NC410 prevents LAIR-1-mediated signaling, we used a reporter cell line that expresses the human LAIR-1 extracellular domain (ECD) fused to CD3ζ, thus conferring positive signaling capacity to LAIR-1 upon interaction with collagen ligands, and an NFAT-GFP reporter to visualize LAIR-1-mediated signal induction (*Lebbink et al., 2006*). Using flow cytometry (*Figure 2E*) and Incucyte microscope imaging (*Figure 2F*, *Figure 2—figure supplement 1*), we observed a dose-dependent inhibition of LAIR-1 NFAT-GFP reporter activity by NC410, indicating that NC410 inhibited collagen-mediated LAIR-1 signaling in a dose-dependent fashion.

## NC410 increases human effector T cell numbers in vivo and controls tumor growth

To study the effect of NC410 on T cell function in a non-tumor model in vivo, we adoptively transferred human peripheral blood mononuclear cells (PBMCs) into NOD scid gamma (NSG) mice. In this model, human xeno-reactive T cells against mouse antigens expand and cause xenogeneic-graft versus-host disease (xeno-GVHD). GVHD models are often used to assess if targeting T cell stimulatory or inhibitory pathways alter T cell activity in vivo with the majority of expanding xeno-reactive T cells acquiring effector or memory phenotypes (*Ehx et al., 2018*). *Figure 3A, B* shows that the majority of human T cells in the spleen on day 6 post transfer acquired an effector memory (EM, CD45RA-CCR7-) phenotype as a percentage of total T cells irrespective of treatment (see *Figure 3—figure supplement 1* for gating strategy). However, it was noted that with increasing doses of NC410, there was a trend of effector T cell (CD45RA+CCR7-) expansion as a percentage of total T cells (*Figure 3B*). In the spleen, it was found that NC410 significantly increased the number of human CD4+EM T cells and human CD8+ effector T cells in a dose-dependent fashion (*Figure 3C–E*).

To evaluate if the increase in xeno-reactive human effector T cell numbers induced by NC410 was associated with increased T cell effector activity, we tested whether NC410 treatment resulted in

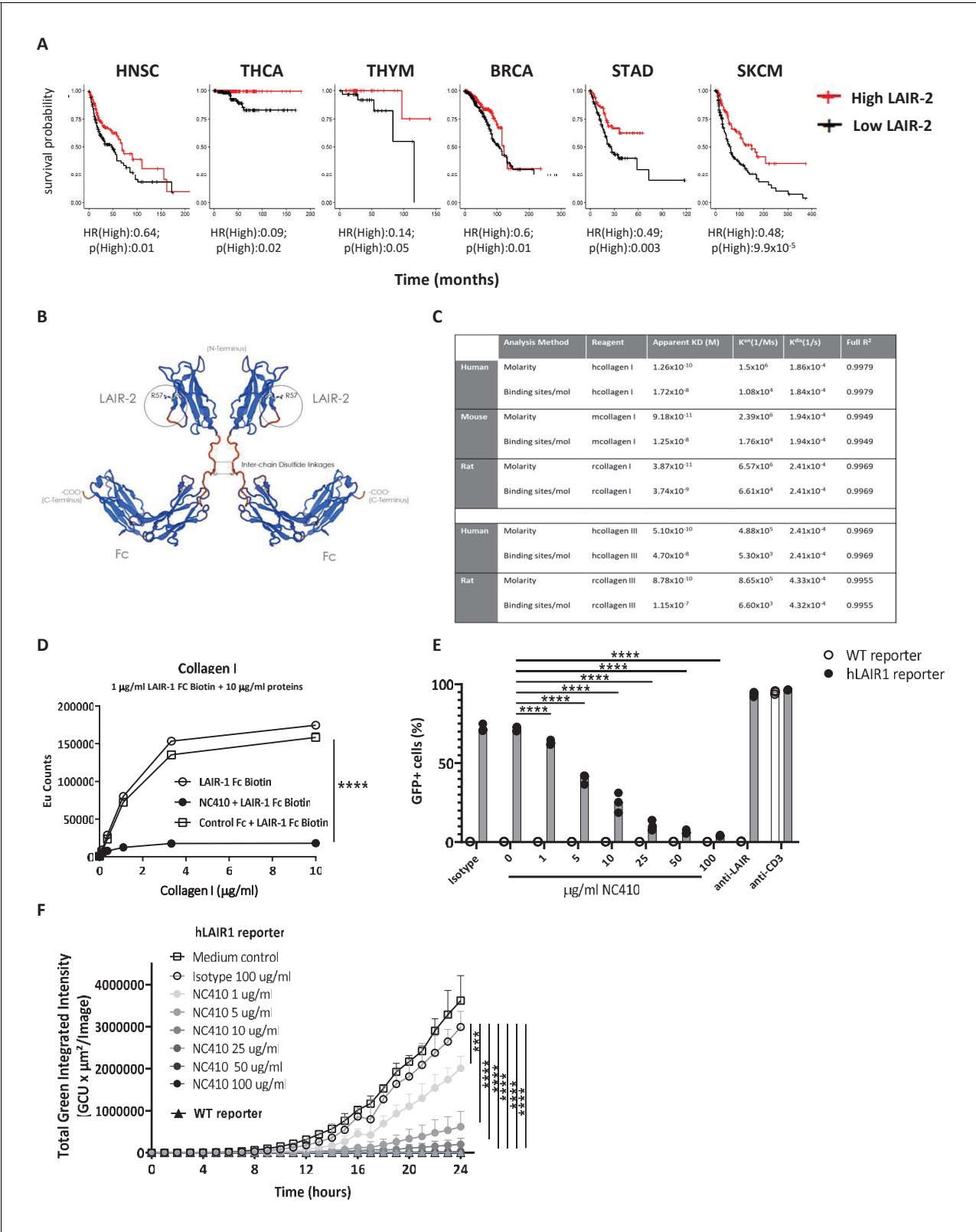

**Figure 2.** Development of NC410, a LAIR-2 Fc protein that blocks LAIR-1-collagen interaction. (**A**) LAIR-2 overexpression is associated with improved overall survival in some tumors: head and neck squamous cell carcinoma (HNSC), thyroid carcinoma (THCA), thymoma (THYM) and skin cutaneous melanoma (SKCM). Patients were grouped in low 25% quantile (black) and high 25% quantile (red) of LAIR-2 mRNA expression for overall survival analysis. Hazards ratio indicating if high expression is associated with poor survival (HR(High)), and p-value (p(High)) indicating the significance of

*Figure 2 continued on next page*

*Figure 2 continued*

association was determined using Wald test as indicated. (B) NC410 is a biologic fusing LAIR-2 with a functional IgG1 to generate a dimeric fusion protein. (C) Avidity characterization of NC410 to human, mouse and rat collagen I and III as measured by Octet analysis. (D) Indicated amounts of collagen I were plate coated, and the binding of soluble LAIR-1 was inhibited by NC410. Asterisks indicate statistical significance (****p<0.0001, two-way ANOVA). (E, F) The human LAIR-1 (hLAIR-1) extracellular domain was fused with CD3z and stably expressed in a cell line containing an NFAT-GFP pathway reporter. LAIR-1 ligation and CD3 ligation induce NFAT-GFP signaling. A parental cell line containing the CD3 NFAT-GFP reporter without LAIR-1 was used as control (WT). NC410 protein was added at increasing concentrations and inhibited human collagen I (5 µg/mL)-mediated NFAT-GFP signaling through LAIR-1 binding by (E) FACS analysis and (F) Incucyte microscopy. Total green integrated intensity of WT and hLAIR-1 reporter cells is shown over time. Points represent the median of n = 3 (with experimental triplicates in each independently performed experiment), and the whiskers indicate the 95% confidence interval (CI). Isotype control was used at the highest concentration (100 µg/mL) and showed no inhibition of NFAT-GFP signaling. Anti-human LAIR (8A8 clone) and anti-mouse CD3 were used as positive controls. Closed circles in (F) indicate NC410 treatment, and open circles indicate control treatment. Significant differences between different treatment groups of hLAIR-1 reporter cells are indicated (and tested using a two-way ANOVA with Dunnett's correction). In all plots: *p≤0.05, **p≤0.01, ***p≤0.001, ****p≤0.0001.

The online version of this article includes the following figure supplement(s) for figure 2:

**Figure supplement 1.** NC410 blocks LAIR-1 functional interactions with collagen.

reduced growth of P815, a mouse (xeno) tumor cell line that does not express collagens to avoid antibody-dependent cellular cytotoxicity (ADCC)-mediated tumor clearance (*Figure 4A, B*). In this model, human PBMCs were injected intravenously on day 0, and P815 cells suspended in Matrigel to provide an ECM-like environment were injected subcutaneously a day later. NC410 or control Fc protein was administered on days 1 and 3 at 1, 3 and 10 mg/kg for comparison to control Fc at 10 mg/kg and a PBS control. Treatment resulted in significantly reduced tumor growth of P815 in the 1 mg/kg treatment group, but not the 3 and 10 mg/kg treatment groups (*Figure 4C*).

To assess the unexpected finding that a lower dose of NC410 was more effective than higher doses, the human T cell response was assessed in this model. Blood was analyzed on days 6 and 13 for changes in CD8$^+$ and CD4$^+$ T cell counts per mL blood (*Figure 4D–G*). NC410 promoted the expansion of both human CD8$^+$ and CD4$^+$ T cells on days 6 and 13. Interestingly, the kinetics of T cell expansion and/or retention changed with dose level. NC410 at 10 mg/kg resulted in a greater increase in T cell numbers on day 6 (*Figure 4D, F*), while 1 mg/kg resulted in the highest T cell numbers on day 13 (*Figure 4E, G*). Together, these results suggested that the sustained expansion of human T cells with a lower dose of NC410 correlated with control of tumor growth.

## NC410 promotes T cell-mediated anti-tumor immunity in a humanized tumor model

To determine if the T cell-promoting and anti-tumor effects observed in the xenogeneic model correlated with the capacity of NC410 to elicit T cell anti-tumor activity against a human tumor, we developed a humanized subcutaneous tumor model using the human HT-29 colorectal tumor cell line. NSG mice were intravenously injected (i.v.) with human PBMCs, and 1 day later were subcutaneously injected (s.c.) with HT-29 cells in Matrigel (*Figure 5A*). NC410 or control treatments began on the same day as tumor implantation. NC410 treatment at 10 mg/kg significantly reduced tumor growth compared to isotype control (*Figure 5A*). In a follow-up dose-titration experiment, it was found that NC410 at 10 mg/kg appeared optimal in the HT-29 model (*Figure 5B*).

Because HT-29 cells also express collagens (*Figure 3—figure supplements 2* and *3*), we investigated whether binding of NC410 to HT-29 cells elicits ADCC. We performed in vitro cytotoxicity assays with HT-29 cells using human PBMC as a source of effector cells (*Figure 3—figure supplement 4*). In an in vitro chromium release cytotoxicity assay, no differences in HT-29 killing by PBMC were observed between NC410, isotype and medium-treated samples (*Figure 3—figure supplement 3F*), suggesting that the anti-tumor effect of NC410 is not mediated by ADCC. In agreement, the therapeutic effect of NC410 was abrogated when we transferred T cell-depleted PBMC into NSG mice (*Figure 3—figure supplement 4*). Thus, T cells are required for the anti-tumor effect mediated by NC410, arguing against ADCC as the main effector mechanism in vivo.

In the same experimental setup, we also compared NC410 to a LAIR-2 fusion protein with a mutated (FES) IgG1 Fc with little or no binding to human and mouse Fc receptors (LAIR-2-FES), which resulted in a loss of therapeutic effect (*Figure 3—figure supplement 4*). Thus, both T cells and a functional Fc were required for NC410 anti-tumor activity and decreased tumor growth.

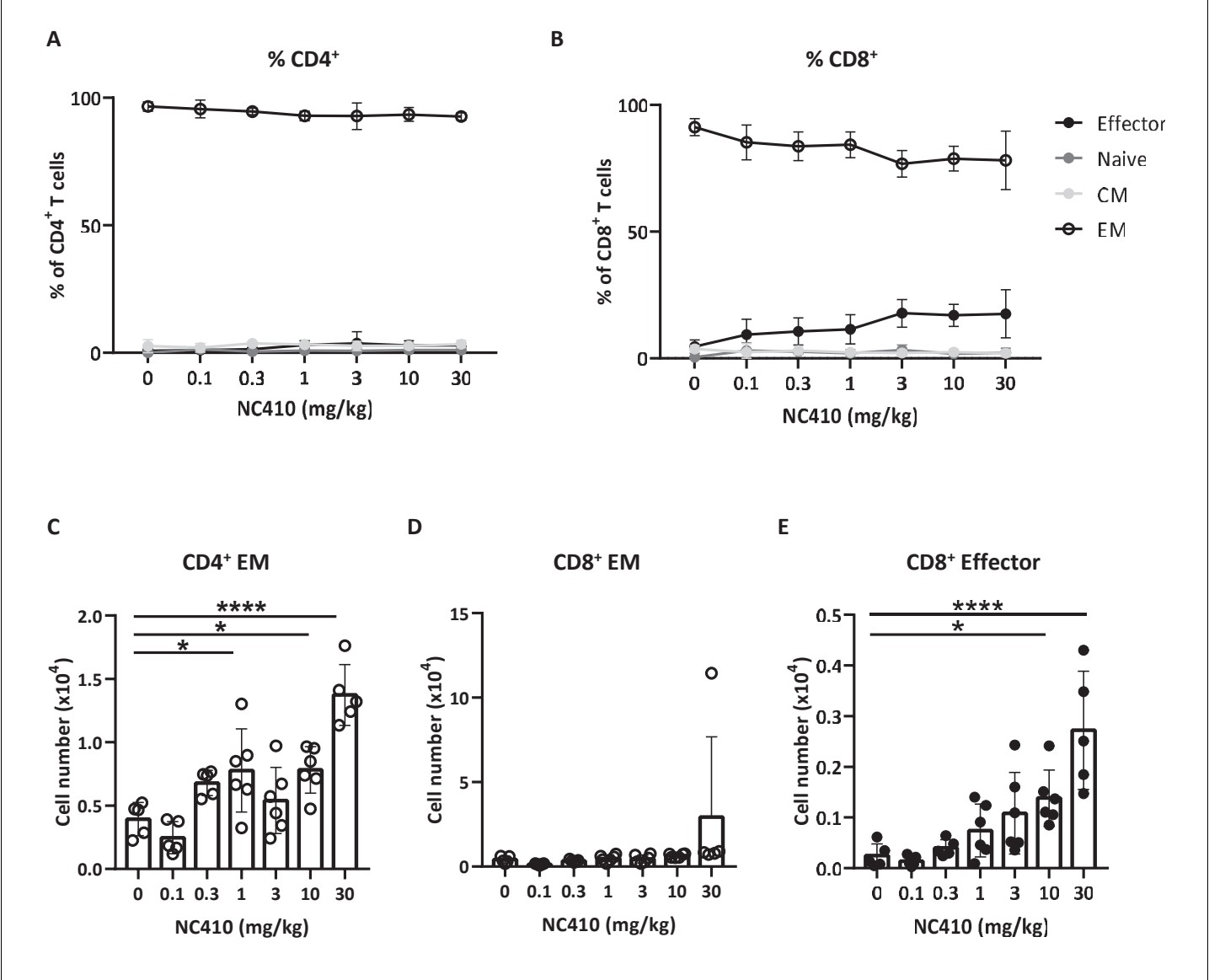

**Figure 3.** NC410 therapy promotes human T cell expansion in a xenogeneic-graft versus-host disease model. In a non-tumor model, $1 \times 10^7$ total human peripheral blood mononuclear cells were adoptively transferred intravenously to NSG mice (N = 6/group) on day 0. Mice were treated with indicated doses of NC410 by intravenous injection on days 0 and 2. On day 6, mice were euthanized and spleens were analyzed for naïve (CD45RA+CCR7+), central memory (CM, CD45RA-CCR7+), effector memory (EM, CD45RA-CCR7-) and effector (CD45RA+CCR7-) CD4+ (A) and CD8+ (B) T cell populations. The graph shows the percentage of T cell subpopulations as a percentage of total human T cells. (C–E) Cell counts of (C) CD4+EM, (D) CD8+EM and (E) CD8+ effector T cells in the spleen. The graphs show the means ± SD (error bars). Asterisks indicate statistical significance: *p<0.05, **p<0.01, ***p<0.001, ****p<0.0001, one-way ANOVA followed by Tukey's multiple comparisons.

The online version of this article includes the following figure supplement(s) for figure 3:

**Figure supplement 1.** Gating strategy to identify human naïve, memory and effector memory T cell subsets in an NSG non-tumor mouse model.

**Figure supplement 2.** HT-29 mRNA collagen expression by RNA sequencing.

**Figure supplement 3.** NC410 binds to collagens on HT-29 cells but does not induce antibody-dependent cellular cytotoxicity.

**Figure supplement 4.** NC410 anti-tumor activity is dependent on T cells and an active IgG1 Fc.

## NC410 promotes infiltration and localized activity of T cells in the TME

Cytokines and chemokines mediate the host response to tumors by activating and directing the trafficking of immune cells into the TME (*Strieter, 2001*; *Waldmann, 2018*). To determine if NC410 promoted infiltration and localized activity of T cells in the TME, immune profiling of HT-29 tumors and

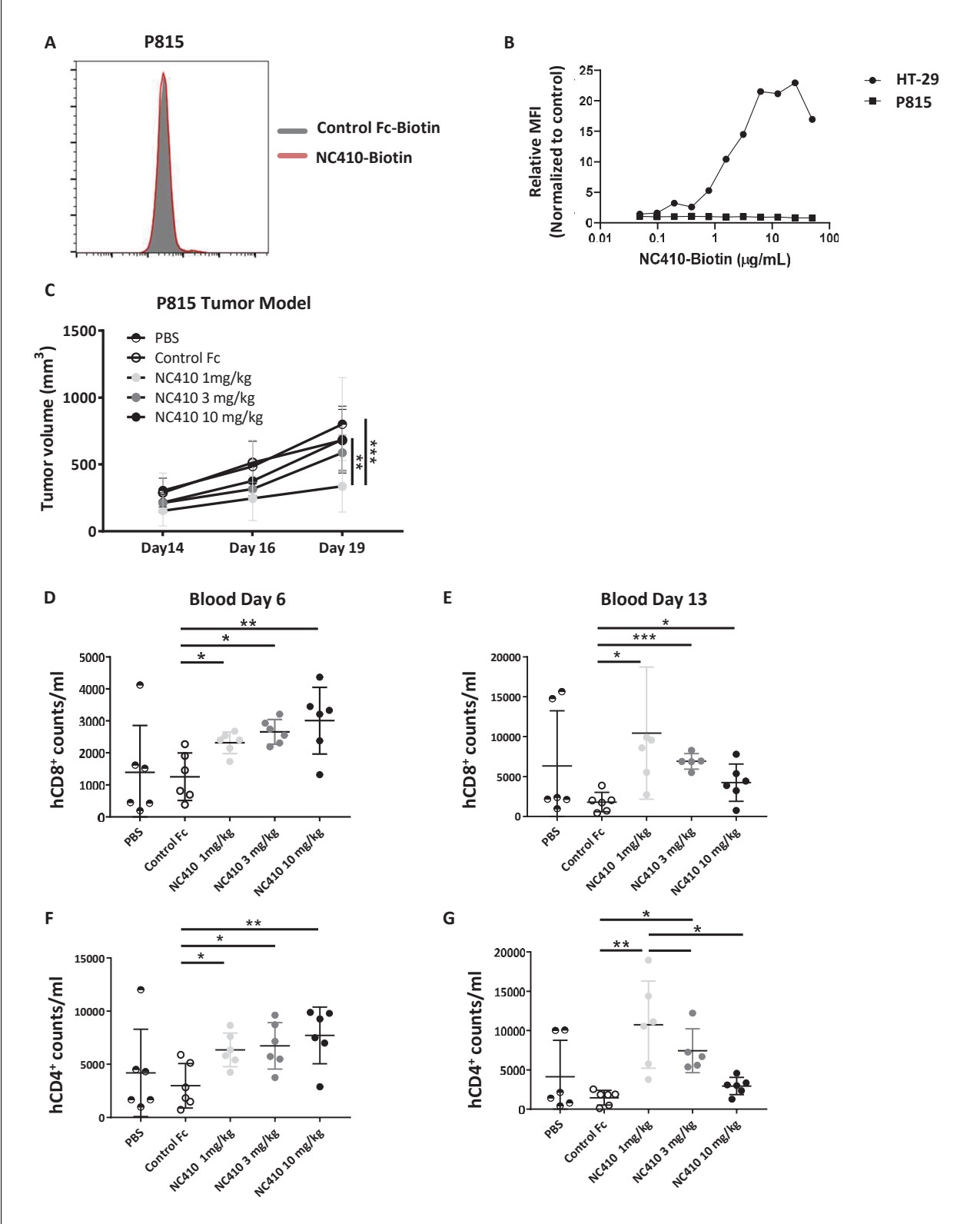

**Figure 4.** NC410 therapy controls in vivo tumor growth of tumor cells that do not express collagen. (**A**) Histogram plot showing mouse P815 tumor cells stained with NC410-biotin or control Fc-biotin. (**B**) Mouse P815 cells and human HT-29 cells were stained with titrated concentrations of NC410-biotin. (**C**) $1.5 \times 10^7$ total human peripheral blood mononuclear cells were adoptively transferred by intravenous injection to NSG mouse on day 0, and $2 \times 10^5$ P815 tumor cells in Matrigel were injected subcutaneously on day 1. Mice were treated on days 1 and 3 with indicated dose of NC410 or control IgG1

*Figure 4 continued on next page*

*Figure 4 continued*

(10 mg/kg) by intraperitoneal injection (N = 6/group). P815 tumor was measured every 2–3 days with a caliper, and tumor volume was calculated. Asterisks indicate statistical significance: **p<0.01, ***p<0.001, two-way ANOVA with Sidak multiple comparisons correction. (**D–G**) CD8$^+$ (**D, E**) and CD4$^+$ (**F, G**) T cell analysis of cell numbers in blood on days 6 (**D, F**) and 13 (**E, G**). The graphs show the means ± SD (error bars). Asterisks indicate statistical significance: *p<0.05 **p<0.01 ***p<0.001, two-way ANOVA with Sidak multiple comparisons correction.

spleen tissues was performed on day 27 after treatment (10 mg/kg dose). HT-29 tumors were removed from euthanized mice, weighed and dissociated for analysis of T cells and soluble factors. For equal comparison of systemic response, the spleen was harvested by identical means for analysis. Analysis of T cell numbers within the tumor showed a significant increase in the number of human CD4$^+$ and a trend towards increased CD8$^+$ T cell numbers in tumors treated with NC410 compared to control (*Figure 5C, D*). To determine the effector capacity of tumor-infiltrating T cells (TILs) from NC410-treated mice, we re-stimulated TILs from digested tumors with anti-CD3 plus anti-CD28 for 5 hr and performed intracellular staining to examine IFN-γ and TNF-α production (*Figure 5E*). After re-stimulation, a significant increase in IFN-γ+ and IFN-γ$^+$TNF-α$^+$ double-positive cells was observed in the NC410 treatment group compared to control. Based on this observation, we further assessed cytokines and chemokines in the local TME and peripheral spleen (*Figure 5F–I*). Analysis of cytokines indicated that IFN-γ and granzyme B were significantly increased in the TME, but not soluble CD40L (*Figure 5F*). Increased expression of CD40L and granzyme B was observed in the spleen (*Figure 5G*). Chemokine analysis indicated that CXCL10, CXCL11 and CXCL12 were all significantly increased in the TME (*Figure 5H*), and significantly decreased in the spleen (*Figure 5I*). Importantly, the concentration of all three chemokines correlated with tumor reduction (*Figure 5J*). These results support a role for NC410 in enhancing the recruitment of T cells and liberating their effector function in the TME. Thus, we established that NC410 has the capacity to induce T cell expansion, T cell tumor infiltration and reduction of tumor growth in vivo.

## NC410 induces specific collagen degradation products

Recently, it has been proposed that specific collagen degradation products (CDPs) reflecting the turnover of specific collagens may be used as biomarkers to non-invasively (in serum) interrogate cell reactivity in the TME and predict response to treatment (*Jensen et al., 2018*; *Jensen et al., 2020*; *Wang et al., 2021*). Because NC410 both engages collagens and promotes local T cell responses, it was surmised that NC410 treatment may result in changes in specific CDP levels, which potentially be indicative of tumor collagen remodeling. Therefore, we investigated whether NC410 treatment would modulate specific CDP levels in the serum of HT-29 tumor-bearing mice transplanted with human PBMC over the course of tumor growth and rejection mediated by NC410 (*Figure 6A, B*). Nine CDPs were analyzed prior to experiment initiation and during 4 weeks of treatment (*Figure 6C*). Two of the CDPs, namely C6M (a collagen VI MMP-2 CDP) and C4G (a collagen IV granzyme B CDP), were significantly increased at week 4 in the NC410-treated group in comparison to the control group (*Figure 6C*). Interestingly, this increase in serum CDPs was observed at the time of NC410-mediated tumor eradication, suggesting that the CDPs were derived from the tumor as a result of T cell activation and effector function.

## NC410 binds collagen-rich tumors with an immune-excluded phenotype

Immunohistochemical (IHC) analysis of serial tissue sections was performed on several different tumor types (*Figure 7A*). A cross section of tumor types was selected from the TCGA analyses of collagen, LAIR-1 and LAIR-2 mRNA expression and/or survival analyses (*Figures 1* and *2A* and *Figure 1—figure supplement 1*). Sections from 10 patients per tumor type of head and neck squamous cell carcinoma (HNSC), glioblastoma (GBM), melanoma (SKCM), non-small-cell lung carcinoma (NSCLC), high-grade serous ovarian carcinoma (HGSC), pancreatic ductal adenocarcinoma (PDAC) and stomach adenocarcinoma (STAD) were stained for hematoxylin and eosin, Masson's Trichrome, anti-LAIR-1, NC410 and the immune cell markers, CD45, CD3, CD68 and CD163 (*Figure 7B, Figure 7—figure supplement 1*). Glioblastoma tumors displayed a very distinct collagen staining compared to the other tumors (*Figure 7—figure supplement 2*) and therefore were excluded from the statistical analysis. NC410 binding co-localized with Masson's Trichrome-positive collagen areas within the TME and LAIR-1$^+$ immune cells was present in all tumors (*Figure 7C, Figure 7—figure*

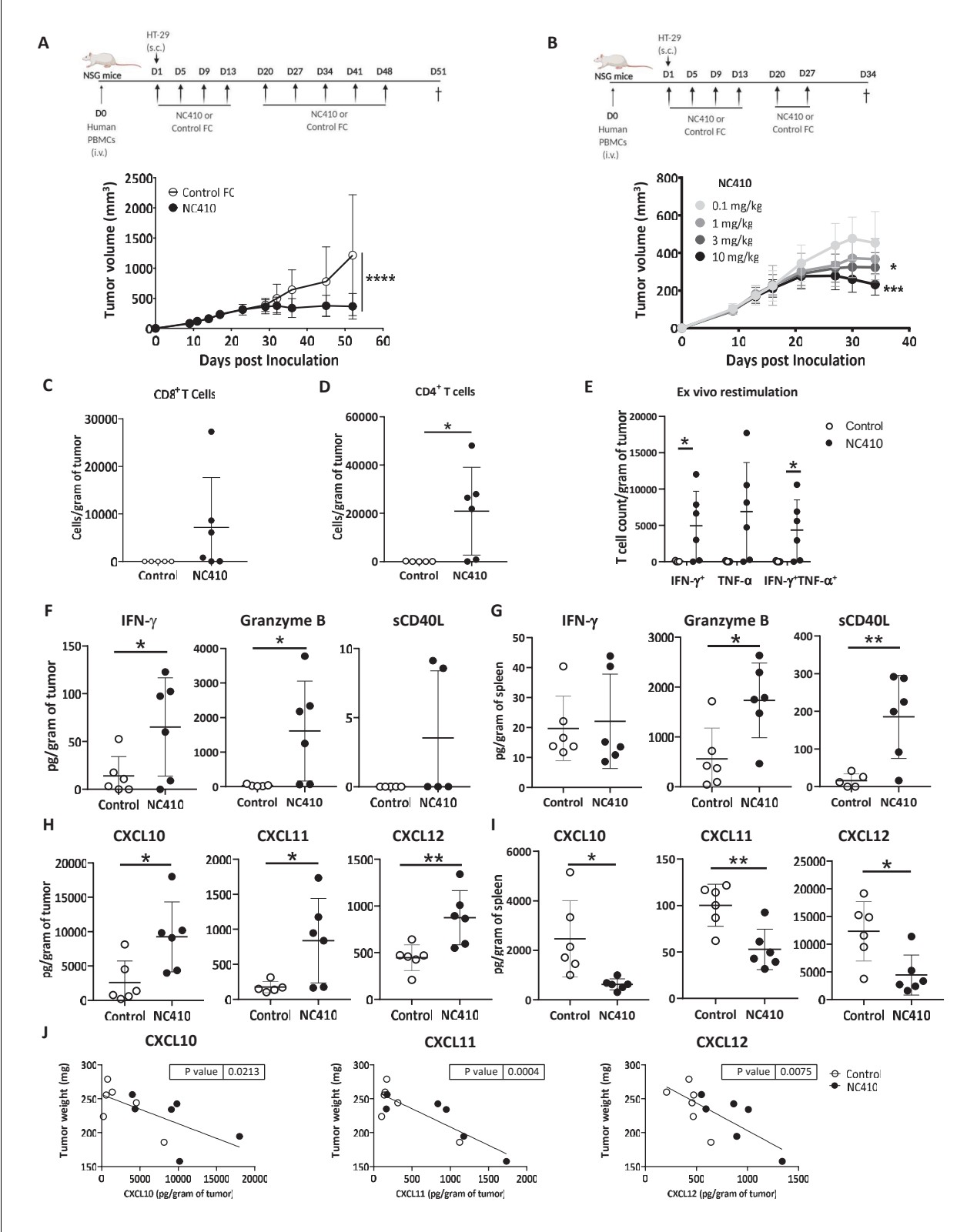

**Figure 5.** NC410 promotes T cell anti-tumor immunity in an HT-29 humanized tumor model. (**A**) Humanized tumor model of HT-29 tumor cells injected subcutaneously in the presence of human peripheral blood mononuclear cells (PBMCs). $2 \times 10^7$ total human PBMCs were adoptively transferred intravenously to NSG mice (N = 6/group) on day 0. $1 \times 10^6$ HT-29 tumor cells were injected subcutaneously with Matrigel on day 1. Mice were treated with 10 mg/kg NC410 or control by intraperitoneal injection, Q4D × 4 doses followed by Q7D until endpoint. Tumor growth was monitored 1–3 times a

*Figure 5 continued on next page*

*Figure 5 continued*

week. Asterisks indicate statistical significance: ****p<0.0001, two-way ANOVA with Sidak multiple comparisons correction. (B) Dose-dependent effect of NC410 (N = 6/group). Experimental conditions are the same as (A), with different doses of NC410. Asterisks indicate statistical significance: *p<0.05, ***p<0.001, two-way ANOVA with Tukey's multiple comparisons correction. (C–G) On day 27 after treatment with 10 mg/mL NC410, tumor and spleen tissues were isolated for tumor-infiltrating T cells (TILs) and cytokine analysis. (C) CD4$^+$ and (D) CD8$^+$ TIL cell numbers in the tumor. The cell number was counted by flow cytometry and normalized to weight (gram) of tumor tissue. Asterisks indicate statistical significance: *p<0.05, two-tailed t-test. (E) Cytokine production by TILs following ex vivo restimulation with phorbol 12-myristate 13-acetate (PMA) and ionomycin for 5 hr. Cells were intracellularly stained for IFN-γ and TNF-α, and the indicated cell populations were counted by flow cytometry and normalized to weight (gram) of tumor tissue. Asterisks indicate statistical significance: *p<0.05, two-tailed t-test. (F–I) Tissue lysate protein was extracted from tumor and spleen tissues. Analysis of tumor (F, H) and spleen (G, I) for cytokines (F, G) and chemokines (H, I) for analysis of local and systemic effects, respectively. Cytokines and chemokines were analyzed by Luminex and presented as the relative levels normalized to weight (gram) of tissue. (J) CXCL10, CXCL11 and CXCL12 correlation with tumor weight. The graphs show the means ± SD (error bars). Asterisks indicate statistical significance: *p<0.05, **p<0.01, two-tailed t-test.

*supplement 2*, healthy tissue *Figure 7—figure supplement 3*). Both myeloid and lymphoid cells within the TME expressed LAIR-1 (*Figure 7—figure supplement 2*). NC410 binding was highest in PDAC in agreement with a collagen-rich TME (*Figure 7C*), and HNSC had the highest number of LAIR-1$^+$ cells (*Figure 7D*). Importantly, LAIR-1$^+$ cells were enriched in NC410-positive areas (*Figure 7E*), suggesting that LAIR-1$^+$ cells were entrapped in the collagen matrix. Most human solid tumors exhibit distinct immunological phenotypes being divided into immune-inflamed, immune-excluded and immune-desert tumors on the basis of immune cell infiltrate and localization (*Chen and Mellman, 2013*; *Binnewies et al., 2018*). By characterizing our cohort according to these immunological phenotypes, we observed that NC410 binding was particularly increased in immune-excluded tumors (*Figure 7—figure supplement 4*). Thus, in immune-excluded tumors, LAIR-1$^+$ cells may be sequestered in collagen-rich areas where NC410 can bind and disrupt LAIR-1-mediated inhibition and adhesion.

In summary, this study provides support for utilizing NC410, a novel LAIR-2 Fc fusion protein that promotes T cell activity, as a supplemental therapy to induce anti-tumor immunity in collagen-rich, immune-excluded tumors in which excluded immune cells express high levels of LAIR-1. A first-in-human phase I clinical trial is currently ongoing to test the safety and tolerability of NC410.

## Discussion

Tumor escape from immune surveillance is a well-established principle (*Beatty and Gladney, 2015*). Tumor progression is accompanied by extensive remodeling of the ECM, leading to the formation of a tumor-specific ECM, which is often more collagen-rich and of increased stiffness (*Cox and Erler, 2014*; *Pickup et al., 2014*). Collagen expression and density have been shown to be associated with a worse prognosis either by directly promoting tumor growth or by preventing immune cell infiltration (*Lu et al., 2012*). Therefore, developing therapeutics that have the capability of targeting collagen-rich tumors is an attractive concept.

It has been demonstrated that collagens can directly modulate T cell function (*Salmon et al., 2012*; *Hartmann et al., 2014*; *Kuczek et al., 2019*), for instance, through the inhibitory collagen receptor LAIR-1 (*Rygiel et al., 2011*; *Park et al., 2020*). We hypothesized that alterations in expression and remodeling of collagen in the TME regulate the threshold of T cell activation through LAIR-1 and are employed by tumor cells to escape immune surveillance. We showed by RNA analysis that high expression of total collagens, LAIR-1, and combined high total collagen and high LAIR-1 are associated with a worse prognosis in multiple tumor types, supporting the notion of disrupting LAIR-1-collagen interactions as a novel immunotherapeutic strategy.

We then hypothesized that LAIR-2 could be utilized as a therapeutic in humans to dually target tumoral collagens and reverse immune suppression. LAIR-2 can be found in vivo in the urine from pregnant women and in fluids such as synovial fluid of rheumatoid arthritis (RA) and osteoarthritis (OA) patients and the circulation of patients with autoimmune thyroid disease ( *Lebbink et al., 2008*; *Olde Nordkamp et al., 2011*; *Simone et al., 2013*). The amount of LAIR-2 protein seems to be related to inflammation (*Olde Nordkamp et al., 2011*). In vitro stimulation of sorted peripheral blood cells revealed that CD4$^+$ T cells are the main producers of LAIR-2 (*Olde Nordkamp et al., 2011*). However, this does not exclude that other cell types might produce

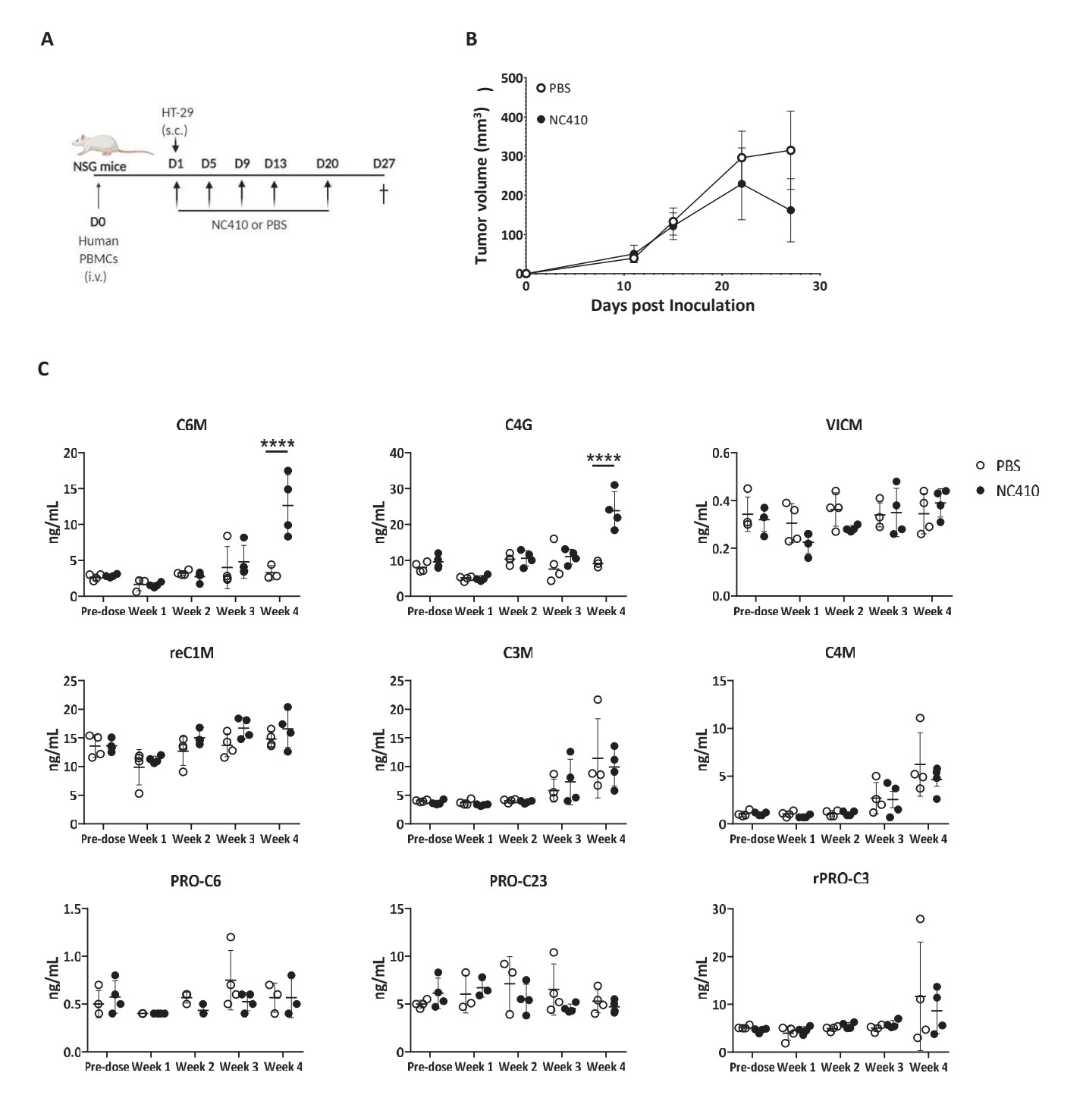

**Figure 6.** NC410 increases collagen degradation products that correlate with tumor regression. (**A**) Schematic representation of the humanized murine tumor model used. HT-29 tumor was injected subcutaneously in the presence of human peripheral blood mononuclear cells. Mice were treated with NC410 or control by intraperitoneal injection, Q4D × 4 doses followed by Q7D until endpoint. Mice were bled prior to start of experiment and weekly for 4 weeks. (**B**) Tumor growth kinetics with NC410. Asterisks indicate statistical significance: *p<0.05, ****p<0.0001, two-way ANOVA with Sidak multiple comparisons. (**C**) Analysis of collagen degradation products in serum at baseline, at weeks 1, 2, 3 and 4. reC1M: neo-epitope of MMP-2,9,13-mediated degradation of type I collagen; C3M: type III collagen degradation by MMP; C4M: type IV collagen degradation by MMP; C6M: neo-epitope of MMP-2-mediated degradation of type VI collagen; PRO-C3: pro-peptide of type III collagen/ECM formation/fibroblast activity; PRO-C6: pro-peptide of type VI collagen; VICM: neo-epitope of MMP-2,8, trypsin-mediated degradation of citrullinated vimentin; C4G: type IV collagen degraded by

*Figure 6 continued on next page*

*Figure 6 continued*

granzyme B (T cell activity/infiltration); closed circles indicate NC410 treatment, and open circles indicate control treatment. The graphs show the means ± SD (error bars). Asterisks indicate statistical significance: ****p<0.0001, two-way ANOVA followed with Sidak multiple comparisons.

LAIR-2 in vivo. Additional studies are required to identify which cells are the main LAIR-2 producers in vivo and the factors that drive its production and secretion. In this study, we showed that targeting and disrupting collagen interactions with LAIR-1 by means of a dimeric LAIR-2 hIgG1 Fc fusion protein, NC410, promoted in vivo T cell activity and had a therapeutic effect in tumor models, which was dependent on T cells. It is interesting to note that we observed peak activity of NC410 at different doses in the P815 (1 mg/kg) vs. HT-29 (10 mg/kg) tumor model. This may be due to the use of different donor PBMCs in these models, use of murine (P815) vs. human (HT-29) tumor cell lines in NSG mice or different mechanisms of action in the two models.

Because many tumors can produce collagens, it is possible that NC410 with an IgG1 Fc could induce ADCC. This was examined in the HT-29 model, which expresses abundant collagens. We did not see enhanced NC410-mediated ADCC in vitro. We also demonstrated the requirement of T cells for anti-tumor activity in our HT-29 model. However, to further address the question of what the Fc portion of NC410 does, we show that LAIR-2 IgG1 protein with a 'silenced' Fc (FES mutation) (*Oganesyan et al., 2008*) loses anti-tumor activity. A recent independently published study (*Xu et al., 2020*) also demonstrates anti-tumor activity with an engineered LAIR-2 Fc protein. In this study, the authors engineered an IgG1 with an N297A to prevent binding to Fc receptors, but further mutated it to include T250Q/M428L mutations to increase binding to neonatal FcR (FcRn), which enhances in vivo bioavailability of the protein. It is possible that the lack of activity from LAIR-2-FES in our study is due to decreased in vivo bioavailability. However, we cannot exclude the possibility that ADCC plays a role in NC410-mediated anti-tumor immunity in some tumor models.

NC410 may also have additional LAIR-1-independent mechanisms of action since we have previously shown that LAIR-2 Fc can efficiently inhibit complement activation via classical and lectin pathways (*Olde Nordkamp et al., 2014a*). Therefore, we cannot exclude that NC410 blocks other receptors or impacts other cells than T cells (*Sivori et al., 2020*) or has additional functions besides outcompeting LAIR-1. These functions of NC410 will be examined as part of ongoing clinical trials, as well as with additional preclinical models.

Humanized mice (Hu-PBMC-NSG) provide an important platform for investigating the modulation of the human immune-mediated anti-tumor responses. While our study demonstrated that the anti-tumor effect promoted by NC410 required T cells, a limitation of the humanized model with adoptive transfer of PBMCs is the absence of human myeloid cells (*Ali et al., 2012*). Therefore, it remains possible that NC410 impacts LAIR-1[+] myeloid function.

Another limitation of our studies is the use of subcutaneous tumor models. Orthotopic models would provide a better comprehensive understanding of how organ-specific ECM, architecture and immune composition would impact response to treatment. Since humanized orthotopic models were not available for this study, we utilized Matrigel as a mimic for ECM.

During ECM turnover, proteolytically cleaved matrix degradation fragments are released into the systemic circulation (*Bonnans et al., 2014*). Serum levels of collagen degradation fragments are elevated in cancer patients compared to healthy controls, and checkpoint blockade alters serum CDP levels (*Jensen et al., 2018*; *Jensen et al., 2020*; *Wang et al., 2021*). NC410 treatment increased specific fragments generated by degradation of collagen VI by MMP-2 (C6M) and collagen IV by granzyme B (C4G). This coincided with a reduction of tumor volume, suggesting that these fragments might be generated by increased infiltration and/or activation of T cells in the TME. Indeed, granzyme B promotes cytotoxic lymphocyte transmigration via basement membrane remodeling of type IV collagen (*Prakash et al., 2014*). We therefore hypothesize that specific granzyme B degraded type IV collagen fragments and other fragments are released to the circulation when T cells infiltrate tumors and may have potential as a clinical pharmacodynamic biomarker. Interestingly, granzyme B-degraded type IV collagen fragments (C4G) associate with favorable anti-CTLA-4 treatment response in metastatic melanoma patients (*Jensen et al., 2018*; *Jensen et al., 2020*; *St-Pierre and Potworowski, 2000*). The potential of C6M and C4G collagen fragments as biomarkers of clinical response will be further investigated in a recently initiated NC410 first-in-human clinical trial (NCT04408599).

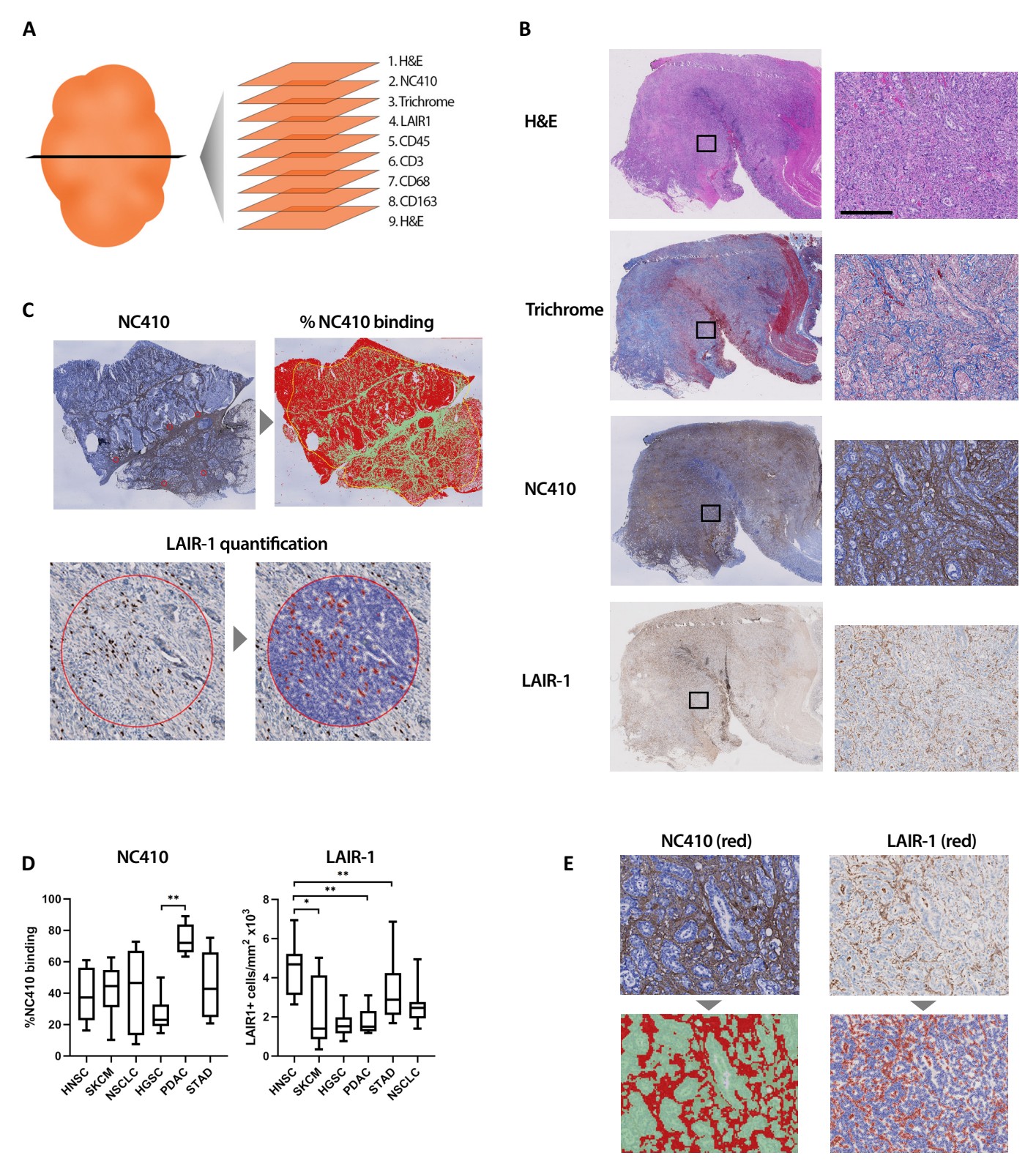

**Figure 7.** Immunohistochemical characterization of LAIR-1$^+$ cells and LAIR-2 Fc binding in primary human tumors. (**A**) Schematic representation of the immunohistochemical stainings performed. (**B**) Representative hematoxylin and eosin (H&E), Masson Trichrome, NC410 and LAIR-1 staining in a stomach tumor specimen. (**C**) Representative NC410 and LAIR-1 analysis. The percentage of NC410 binding to tissue within a tumor was calculated by dividing the NC410$^+$ stained area by the total tumor area. The number of LAIR-1$^+$ cells was calculated by dividing the total number of positive cells

*Figure 7 continued on next page*

*Figure 7 continued*

within five regions of interest (ROIs) by the total surface in mm² of these ROIs. (D) Quantification of LAIR-1 and NC410 staining of 9–10 patients per tumor type across six different tumor types (head and neck squamous cell carcinoma [HNSC], skin cutaneous melanoma [SKCM], non-small cell lung carcinoma [NSCLC], high-grade serous ovarian carcinoma [HGSC], pancreatic adenocarcinoma [PDAC] and stomach adenocarcinoma [STAD]). (E) Higher magnification pictures of stomach cancer specimens show LAIR-1+ cells (depicted in red, right side) co-localizing with NC410-positive areas (depicted in red, left side).

The online version of this article includes the following figure supplement(s) for figure 7:

**Figure supplement 1.** Characterization of LAIR-1+ cells in primary human tumors.
**Figure supplement 2.** Immunohistochemical analysis of primary human tumors.
**Figure supplement 3.** Immunohistochemical analysis of healthy tissues.
**Figure supplement 4.** NC410 preferentially binds to tumors with an immune-excluded phenotype.

NC410 showed decreased tumor growth as a monotherapy, but it might even show better clinical efficacy when used in combination with currently used checkpoint receptor inhibitors such as CTLA4 and/or PD-1. In accordance, it was recently shown that abrogating LAIR-1 immunosuppression through LAIR-2 overexpression or SHP-1 inhibition sensitizes resistant lung tumors to anti-PD-1 therapy (*Peng et al., 2020*).

Circulating LAIR-2 protein concentration is low or undetectable in healthy individuals (*Lebbink et al., 2008*; *Olde Nordkamp et al., 2011*). In the presence of limited endogenous LAIR-2, LAIR-1 can bind collagens, thereby allowing the inhibitory receptor to signal and prevent or reduce T cell activity and/or retain LAIR-1+ cells in collagen-rich areas. In the TME, increased levels of collagens will therefore promote tumor immune escape in the absence of increased LAIR-2. Our analysis of TCGA data indeed revealed that enhanced expression of endogenous LAIR-2 in some tumors associated with better prognosis, suggesting that further increasing LAIR-2 in vivo could have a therapeutic advantage. NC410 has a higher avidity to collagen than endogenous LAIR-2 due to its dimeric structure since in vivo LAIR-2 is expressed as a monomer, enhancing its potential to block the inhibitory capacity of membrane-bound LAIR-1. NC410 binds both healthy and tumoral collagen and theoretically could be sequestered before reaching the tumor site. We hypothesize that the avidity of NC410 towards tumoral collagen is higher than to healthy collagen, therefore resulting in a therapeutic effect, but this needs additional study. Other LAIR-1 ligands, such as C1q, have been shown to be increased in tumors (*Bulla et al., 2016*) and may also provide a local inhibitory effect that could be potentially reversed by NC410.

Understanding LAIR-1 and collagen expression in the context of tumor structure and co-localization could help to identify indications that may best benefit from targeting this axis. Our IHC analyses demonstrated that LAIR-1+ cells were present within NC410-binding collagen-rich areas. This indicated that an abundance of collagen may trap immune cells in the tumor stroma, possibly by binding to LAIR-1. This was most notable in immune-excluded tumors. The immune-excluded phenotype is characterized by the presence of immune cells that cannot penetrate the parenchyma of the tumors but instead are located in the stroma that surrounds the tumor cells (*Chen and Mellman, 2017*). These are indications that currently fare poorly with PD-1/PD-L1 immunotherapy (*Jiang et al., 2018*), and thus NC410 could benefit an entirely new class of cancer patients when used as either a monotherapy or in combination with checkpoint inhibitors. Our data support NC410 as a novel immunomedicine for targeting immune-excluded collagen-rich tumors and enabling normalization of the tumor-immune microenvironment.

## Materials and methods

### Key resources table

| Reagent type (species) or resource | Designation | Source or reference | Identifiers | Additional information |
|---|---|---|---|---|
| Genetic reagent (*Mus musculus*) | NOD-SCID IL2Rγnull | Jackson Labs | Stock #: 005557, RRID: IMSR_JAX:005557 | Female mice |

*Continued on next page*

*Continued*

| Reagent type (species) or resource | Designation | Source or reference | Identifiers | Additional information |
|---|---|---|---|---|
| Biological sample (*Homo sapiens*) | Human PBMCs | In vitro studies – healthy donors in agreement with ethical committee of the University Medical Center Utrecht<br>In vivo studies – Leukopaks (StemCell) | | |
| Cell line (*M. musculus*) | hLAIR-1 reporter cell line | Meyaard Lab | | 2B4 T cell hybridoma cells transduced with a NFAT-GFP reporter and hLAIR-1-CD3ζ |
| Cell line (*M. musculus*) | WT reporter cell line | Meyaard Lab | | 2B4 T cell hybridoma cells transduced with a NFAT-GFP reporter |
| Cell line (*Homo sapiens*) | HT-29 | ATCC | ATCC HTB-38 (RRID: CVCL_0320) | |
| Cell line (*M. musculus*) | P815 | ATCC | ATCC TIB-64 (RRID:CVCL_2154) | |
| Cell line (*Cricetulus griseus*) | CHO cells | Lonza | CHOK1SV, RRID:CVCL_DR97 | CHOK1-SV-GS (glutamine synthase)-KO cells |
| Antibody | Anti-human CD45-BV421 (mouse monoclonal) | ThermoFisher Scientific | Cat number: 50-166-070 | (Flow cytometry: 1:40) |
| Antibody | Anti-mouse CD45-APC (rat monoclonal) | ThermoFisher Scientific | Cat number: 17-0451-82 | (Flow cytometry: 1:200) |
| Antibody | Anti-human CD3-PerCP.Cy5.5 (mouse monoclonal) | ThermoFisher Scientific | Cat number: 45-0037-42 | (Flow cytometry: 1:40) |
| Antibody | Anti-human CD8-AF488 (rat monoclonal) | ThermoFisher Scientific | Cat number: 53-0081-82 | (Flow cytometry: 1:40) |
| Antibody | Anti-human CD4-BV711 (mouse monoclonal) | Biolegend | Cat number: 300558 | (Flow cytometry: 1:40) |
| Antibody | Anti-Human CD45RA-eF450 (mouse monoclonal) | ThermoFisher | Cat number: 48-0458-42 | (Flow cytometry: 1:40) |
| Antibody | Anti-human CCR7-PE (mouse monoclonal) | Biolegend | Cat number: 353204 | (Flow cytometry: 1:40) |
| Antibody | Anti-human TNF-α-PE (mouse monoclonal) | ThermoFisher Scientific | Cat number: 12-7349-41 | (Flow cytometry: 1:40) |
| Antibody | Anti-human IFN-γ-PECy7 (rat monoclonal) | ThermoFisher Scientific | Cat number: 25-7311-41 | (Flow cytometry: 1:40) |
| Antibody | Anti-human LAIR-1 (rabbit polyclonal) | ATLAS antibodies | Cat number: HPA011155 | (Immunohistochemistry: 1:500) |
| Antibody | Anti-human LAIR-1 (mouse monoclonal) | BD Pharmingen | Cat number: 550810 | (Immunohistochemistry: 1:100) |
| Antibody | NC410-biotin | This paper/NextCure | | See Materials and methods (Immunohistochemistry: 1:500) |
| Antibody | Anti-human CD45 (mouse monoclonal) | DAKO | Cat number: GA751 | (Immunohistochemistry: 1:25) |
| Antibody | Anti-human CD3 (rabbit polyclonal) | DAKO | Cat number: GA503 | (Immunohistochemistry: 1:100) |
| Antibody | Anti-human CD68 (mouse monoclonal) | Novocastra | Cat number: NCL-L-CD68 | (Immunohistochemistry: 1:1600) |
| Antibody | Anti-human CD163 (mouse monoclonal) | Novocastra | Cat number: NCL-L-CD163 | (Immunohistochemistry: 1:800) |

*Continued on next page*

*Continued*

| Reagent type (species) or resource | Designation | Source or reference | Identifiers | Additional information |
|---|---|---|---|---|
| Antibody | Pan-collagen antibody (rabbit polyclonal) | ThermoFisher Scientific | Cat number: PA1-85324 | (Immunofluorescence: 1:200) |
| Antibody | Streptavidin APC | eBioscience | Cat number: 17-4317-82 | (Flow cytometry: 1:200) |
| Antibody | IgG1-AlexaFluor 594 | ThermoFisher Scientific | Cat number: A-21125 | (Immunofluorescence: 1:200) |
| Antibody | Streptavidin AlexaFluor 594 | ThermoFisher Scientific | Cat number: S11227 | (Immunofluorescence: 1:2000) |
| Peptide, recombinant protein | LAIR-1 FC | This paper/NextCure | | See Materials and methods LAIR-1- Fc, LAIR-2-Fc (NC410) and LAIR-2-Fc (NC410) FES generation |
| Peptide, recombinant protein | LAIR-2 FC | This paper/NextCure | | See Materials and methods LAIR-1- Fc, LAIR-2-Fc (NC410) and LAIR-2-Fc (NC410) FES generation |
| Peptide, recombinant protein | LAIR-2 FES | This paper/NextCure | | See Materials and methods LAIR-1- Fc, LAIR-2-Fc (NC410) and LAIR-2-Fc (NC410) FES generation |
| Peptide, recombinant protein | Human collagen I | R&D Systems; Stem Cell | Cat number: 6220 CL-020 (R&D); 07005 (StemCell) | |
| Peptide, recombinant protein | Mouse collagen I | Ray Biotech | Cat number: DF-01-0058 | |
| Peptide, recombinant protein | Rat collagen I | Yo Protein | Cat number: ABIN628947 | |
| Peptide, recombinant protein | Human collagen III | R&D Systems | Cat number: 7294 CL-020 | |
| Peptide, recombinant protein | Rat collagen III | Yo Protein | Cat number: ABIN377054 | |
| Commercial assay/kit | mouse tumor dissociation kit | Miltenyi | Cat number: 130-096-730, RRID:SCR_020285 | |
| Commercial assay/kit | Cell Stimulation Cocktail plus protein transport inhibitors | ThermoFisher Scientific | Cat number: 00-4970-93 | |
| Commercial assay/kit | Cytofix/CytoPerm Plus Fixation/Permeabilization Kit | BD | Cat number: 555028 | |
| Commercial assay/kit | Singleplex Luminex Protein Assay Kit | ThermoFisher Scientific | Cat number: EPX010-10420-901 | |
| Commercial assay/kit | Human T cell positive selection kit | StemCell | Cat number: 17851 | |
| Commercial assay/kit | Optiview DAB IHC detection kit | Ventana kit | Cat number: 760–700 | |
| Other | Zombie NIR viability dye | Biolegend | Cat number: 423105 | |
| Other | ACK lysis buffer | KD medical | Cat number: 50-101-9080 | |
| Other | RIPA Lysis buffer | ThermoFisher Scientific | Cat number: 89900 | |
| Other | proteinase inhibitor | Roche | Cat number: 4693116001 | |
| Other | DNase | Millipore | Cat number: 69182–3 | |
| Other | Collagenase (from *Clostridium histolyticum*) | Sigma | Cat number: C0130 | |
| Other | Triton X-100 | Roche | Cat number: 10789704001 | |
| Other | EZ-Link NHS-PEG4-Biotin | ThermoFisher | Cat number: 21329 | |

*Continued on next page*

*Continued*

| Reagent type (species) or resource | Designation | Source or reference | Identifiers | Additional information |
|---|---|---|---|---|
| Other | ZebaSpin Desalting Columns | ThermoFisher | Cat number: 87770 | |
| Other | DELFIA wash buffer | PerkinElmer | Cat number: 1244-114 | |
| Other | DELFIA enhancement solution | PerkinElmer | Cat number: 4001-0010 | |
| Other | Europium-labeled Streptavidin | PerkinElmer | Cat number: 1244-360 | |
| Other | Masson's Trichrome | Abcam | Cat number: Ab150686 | |
| Other | Hematoxylin | Sigma-Aldrich chemie | Cat number: 51275-1L | |
| Other | DAPI VectaShield hardset | Vector Lab | Cat number: H-1500-10, RRID:AB_2336788 | |
| Software, algorithm | QuPath version 0.2.0 | QuPath | RRID:SCR_018257 | |
| Software, algorithm | GraphPad Prism 8.0 | GraphPad Prism | RRID:SCR_002798 | |
| Software, algorithm | | TCGA database (https://cancergenome.nih.gov/) | RRID:SCR_003193 | R codes for TCGA analysis Files can be found in the supplied Source code files |

## Mice

NSG (NOD-SCID IL2Rγnull) female mice were purchased at the age of 6–8 weeks old from Jackson Labs. Upon arrival at NextCure, mice were divided into 5–6 mice per cage and kept in the quarantine room for at least 6 days to acclimate to the environment. All mouse studies were performed at NextCure based on Institutional Animal Care and Use Committee standards according to the protocols of NextCure Animal (NCA) Study 164 (NCA#164 for *Figure 3*), NCA#122 (for *Figure 4*), NCA#209 (for *Figure 5*), NCA#217 (for *Figure 6*) and NCA#270 (for *Figure 3—figure supplement 4*).

## Bioinformatics

For collagen expression, the mRNA expression of 44 collagen genes in normal and tumor tissue for each tumor type was queried using TCGA database (https://cancergenome.nih.gov/). Only those cancers that contained both normal and tumor data were considered for further analyses. mRNA collagen data was available in TCGA for 43/44 collagen genes except for COL6A5. The transcript per million (TPM) values were log2 transformed and averaged per individual across all collagen genes. The distribution of average expression across individuals was plotted using ggplot utility version 2.3.3.2 in R version 4.0.2. The distribution of expression in normal tissues was compared against those in tumor tissue using a non-parametric Wilcoxon test in R.

For overall survival analysis, the log2 transformed collagen expression across 43 collagen genes was averaged in each cancer. The average values were divided into four quantiles, and the patients in lower quantile were considered as individuals with low expression and those in the upper quantile were considered as those with high expression. The estimate of survival based on collagen expression was determined using the Kaplan–Meier method. The survival curves were drawn using ggsurvplot function in the survminer R package. The survival curves for the protein-tumor combination where the p-value was less than 0.05 were considered significant. The expression and overall survival analyses for LAIR-1 and LAIR-2 were performed in the same manner. R codes can be found in the Source code files.

For collagen gene expression analysis of HT-29 cells, data was acquired from the public dataset GSE41586 (*Xu et al., 2013*). Raw count data of untreated HT-29 cells was retrieved and normalized using the DESeq2 package (v1.28.1) in R (v4.0.2). Data was then log2 transformed and plotted using the ggplot2 package (v3.3.2).

## Cells and antibodies

2B4 T cell hybridoma cells transduced with a NFAT-GFP reporter and hLAIR-1-CD3ζ, the hLAIR-1 reporter cells, or transduced with a NFAT-GFP reporter, the WT reporter cells, were cultured in RPMI 1640 (Life Technologies) supplemented with 10% fetal bovine serum (FBS) (Sigma-Aldrich) and 1% penicillin/streptomycin (Gibco). P815 cells (ATCC) were cultured in DMEM, 2 mM L-glutamine, 25 mM HEPES, 10% FBS and Penicillin-Streptomycin (100 U/mL to 100 µg/mL). HT-29 cells (ATCC) were cultured in IMDM, 2 mM L-glutamine, 25 mM HEPES, 10% FBS and Penicillin-Streptomycin (100 U/mL to 100 µg/mL). CHO cells (Lonza) were cultured in CD CHO medium (ThermoFisher Scientific). All cell lines were mycoplasma negative. Anti-human CD45 (hCD45)-BV421, anti-mouse CD45 (mCD45)-APC, anti-human CD3(hCD3)-PerCP.Cy5.5, anti-human CD8(hCD8)-AF488, anti-human CD45RA (hCD45RA)-eF450, anti-human TNF-α (hTNF-α)-PE and anti-human IFN-γ (hIFN-γ)-PECy7 were from ThermoFisher Scientific. Anti-human CD4 (hCD4)-BV711 and anti-human CCR7 (hCCR7)-PE was from Biolegend.

## LAIR-1 Fc, LAIR-2 Fc (NC410) and LAIR-2 FES generation

The human LAIR-1 and LAIR-2 genes were synthesized by GeneArt and genetically fused with the N terminus of IgG1 Fc domain. Stable CHO cell lines expressing recombinant human LAIR-1 Fc, LAIR-2 Fc or LAIR-2 FES (hIgG1 FES mutation silenced Fc receptor binding) fusion protein were developed using the Lonza GS system. Briefly, $5 \times 10^7$ CHO cells were transfected by electroporation using 80 µg of linearized plasmid DNA in a 0.4 cm cuvette. Following electroporation (300 V, 900 µF), cells were resuspended in 100 mL glutamine-free CD CHO medium (ThermoFisher Scientific). The following day, MSX (Millipore) was added to a final concentration of 50 µM, and cells were monitored for the next two weeks as prototrophic cells began to grow. Single clone of stably transfected cells was cultured, and supernatant was harvested and purified by affinity chromatography. The protein purity was determined by HPLC and sodium dodecyl sulfate-polyacrylamide gel electrophoresis. For the LAIR-1-collagen binding assay and NC410 flow cytometry, LAIR-1 Fc and NC410 were biotinylated with EZ-Link NHS-PEG4-Biotin (ThermoFisher Scientific) and free biotin was removed by ZebaSpin Desalting Columns (ThermoFisher Scientific) following manufacturer's instructions.

## Human PBMC preparation for in vitro experiments

PBMCs were isolated from blood of healthy donors (in agreement with ethical committee of the University Medical Center Utrecht [UMCU] and after written informed consent from the subjects in accordance with the Declaration of Helsinki) using standard Ficoll density gradient centrifugation. Briefly, blood was diluted 1:1 with PBS and layered on top of 15 mL of Ficoll-Paque (GE healthcare) in 50 mL conical tube. Suspension was centrifuged at 400 × g for 20 min at 20°C in a swinging bucket rotor without brake. The mononuclear cell layer at the interphase was carefully collected and transferred to a new 50 mL conical tube. The cells were washed with PBS and centrifuged at 300 × g for 10 min at 20°C. The supernatant was discarded, and the cell pellet was washed twice with 50 mL PBS. The isolated PBMCs were immediately used for in vitro studies.

## Binding and blocking studies

### Octet avidity analysis

A ForteBio Octet RED96 instrument was used for avidity assessments. The anti-human Fc antibody capture sensors (ForteBio) were first loaded with LAIR-2-Fc followed by an association step where the loaded sensor was dipped into wells containing human, mouse or rat collagen I (human, R&D Systems; mouse, Ray Biotech; rat, Yo Protein) or collagen III (human, R&D Systems; rat, Yo Protein). NC410 protein was diluted in assay buffer (ForteBio) at 20 µg/mL, and the collagen concentration ranged from 1.56 µg/mL to 100 µg/mL for collagen I and from 0.78 µg/mL to 50 µg/mL for collagen III. Data processing was conducted using the Octet's Data Analysis 9.0 software.

### Time-resolved fluorometry (TRF) immunoassay

Enzyme Immuno Assay (EIA) plates were coated with human collagen I (StemCell) in 0.01 N HCL (100 µL/well) overnight at 4°C. The following day, plates were equilibrated to ambient temperature and washed three times (300 µL/well) with DELFIA Wash buffer (PerkinElmer). The plates were blocked for non-specific binding with 3% BSA (200 µL/well, Millipore) for 1 hr. Plates were washed

three times (300 µL/well) with DELFIA wash buffer, and an NC410-biotin and human LAIR-1 Fc mixture (50 µL/well) was added to plates and incubated for 2 hr at ambient temperature. The plates were washed three times (300 µL/well) with DELFIA wash buffer. Europium-labeled Streptavidin (Eu-SA) (100 µL/well, PerkinElmer) was diluted 1:1000 in DELFIA assay buffer and was added to plates and incubated for 1 hr at ambient temperature. Following the incubation, the plates were washed with 300 µL/well of DELFIA wash buffer. DELFIA enhancement solution was equilibrated to ambient solution during detection antibody incubation. Following the last wash, 100 µL of DELFIA enhancement solution (PerkinElmer) was added to each well and incubated on a plate shaker for 5 min prior to reading on an EnVision plate reader with excitation at 340 nm, and fluorescence reading at 615 nm (PerkinElmer).

## Flow cytometry blocking studies

To assess the blocking capacity of NC410, a titration assay with LAIR-1 reporter cells was performed as previously described (*Lebbink et al., 2006*). Black Falcon clear flat bottom 96-well plates were coated with 5 µg/mL human collagen I (Sigma-Aldrich) in 2 mM acetic acid (Merck), anti-mouse-CD3 (BD), anti-human-LAIR-1 antibody (clone 8A8) in PBS (Sigma-Aldrich) or isotype control (eBioscience) in PBS by spinning down for 3 min at 1700 rpm and incubating overnight at 4°C. The next day, plates were washed with PBS and pre-incubated with the indicated concentrations of NC410 or isotype control (NextCure) in culture medium by spinning down for 5 min at 1500 rpm at room temperature (RT) and incubating for 2 hr at 37°C.

WT and hLAIR-1 reporter cells were harvested and seeded at $1 \times 10^6$ cells/mL in 50 µL/well on top of the collagen and fusion protein-treated wells and spun down for 3 min at 1700 rpm at RT. Plates were incubated overnight, approximately 16 hr, at 37°C, and GFP expression was measured on a LSRFortessa (BD Biosciences).

## Incucyte assay

Plates were prepared similarly to the flow cytometric analysis. After adding reporter cells to collagen and fusion protein-treated wells, plates were placed in the Incucyte S3 (Sartorius) and green fluorescence of the GFP expressed by the reporter cells was imaged every hour for 24 hr.

Analysis of Incucyte images was performed using the Incucyte 2020A analysis program (Sartorius), where green fluorescence was evaluated using Top-Hat segmentation (radius 100 µm and threshold 2 GCU), edge split turned on, minimum mean intensity of 3 GCU and an area filter of 600 $µm^2$ to calculate the total green integrated intensity (GCU $\times µm^2$/image) per well.

## In vivo experiments

Leukopaks (StemCell) were diluted with PBS and layered with 35 mL of diluted cell suspension over 15 mL of Ficoll-Paque (GE healthcare) in 50 mL conical tube. Suspension was centrifuged at $400 \times$ g for 30 min at 20°C in a swinging bucket rotor without brake. The mononuclear cell layer at the interphase was carefully collected and transferred to a new 50 mL conical tube. The cells were washed with PBS and centrifuged at $300 \times$ g for 10 min at 20°C. The supernatant was discarded, and the cell pellet was washed twice with 50 mL PBS. The isolated PBMCs were frozen and stored in liquid nitrogen.

Prior to in vivo studies, PBMCs were rested overnight in RPMI 1640, 2 mM L-glutamine, 10 mM HEPES, 10% FBS, Penicillin-Streptomycin (100 U/mL to 100 µg/mL) and 250 U/mL DNase (Millipore). Female NSG mice were injected intravenously (i.v.) with $1–2 \times 10^7$ PBMC or T cell-depleted (StemCell human T cell-positive selection kit) PBMCs in 100 µL of 1× PBS. The next day, $2 \times 10^5$ P815 cells in PBS, or $0.1–1 \times 10^6$ HT-29 cells in PBS with 50% Matrigel (Corning) were injected subcutaneously on the right flank. Mice were randomly assigned into treatment or control groups (six mice per group). The sample size per group was determined with resource equation approach n = DF/k + 1, where n = number of samples per group, k = number of groups and DF = degrees of freedom with acceptable range between 10 and 20 in ANOVA and t-test (*Charan and Kantharia, 2013*; *Arifin and Zahiruddin, 2017*). Beginning on day 1, LAIR-2 Fc and control protein were injected intraperitoneally (i.p.) Q4D × 4 doses followed by Q7D until the endpoint. Tumor size was monitored 2–3 times a week. Tumor volumes were determined according to the formula tumor volume = 0.5 × (shorter diameter)$^2$× longer diameter. At endpoint, tumor and spleen tissues were collected for T

cell population and/or cytokine analysis. In some studies, blood was collected weekly for T cell population and collagen degradation analysis. Mice were randomized in all tumor models prior to treatment. Tumor measurements were performed in a blinded manner.

For ex vivo analysis, cells were stained with Zombie NIR viability dye (Biolegend) in PBS at RT for 10 min. After washing with FACS buffer (2% FBS in PBS), cells were stained with antibodies against cell surface antigens at 4°C for 30 min. For the intracellular staining, cells were stimulated with Cell Stimulation Cocktail plus protein transport inhibitors (ThermoFisher Scientific) at 37°C for 5 hr followed by Zombie NIR and cell surface antigen staining. After cell fixation and permeabilization, the intracellular TNF-$\alpha$ and IFN-$\gamma$ were stained following the instructions of BD Cytofix/CytoPerm Plus Fixation/Permeabilization Kit. All antibodies were used at the concentrations recommended by manufacturers. Stained cells were washed and resuspended in 150 µL FACS buffer, and 80 µL of samples were acquired on an Attune flow cytometer (ThermoFisher Scientific). Human CD4$^+$ and CD8$^+$ T cells were gated based on live/hCD45$^+$mCD45$^-$hCD3$^+$hCD4$^+$hCD8$^-$ and live/hCD45$^+$mCD45$^-$hCD3$^+$hCD4$^-$hCD8$^+$, respectively.

To prepare mouse blood T cells for staining, 80–200 µL blood was treated with 3 mL ACK lysis buffer (KD medical) to lyse the red blood cells for 5 min at RT followed by washing with PBS. The initial volume of blood was recorded for calculation of cell counts per mL of blood according to the formula: Cell counts per mL of blood = [acquired counts $\times$ 150 $\times$ 1000] ÷ [initial blood volume (µL) $\times$ 80].

To prepare the single-cell population from tumors for staining, tumor tissues were weighed, cut into small pieces and digested with mouse tumor dissociation kit (Miltenyi) and dissociated with gentleMACS Dissociator (Miltenyi). The tumor weight was recorded for the normalization of cell counts.

To assess cytokines from tissues, tumor and spleen tissues were weighed and cut into small pieces in 1.5 mL Eppendorf tube on ice. 200 µL of RIPA Lysis buffer (ThermoFisher Scientific) was added with proteinase inhibitor (Roche) and 250 U/mL of DNase (Millipore), and the tissues were dissociated with pellet pestles (Sigma) on ice. The samples were kept on ice for 30 min, vortexing occasionally. Centrifuge at 10,000 $\times$ g for 20 min at 4°C to pellet cell debris and then transfer the supernatant to a fresh Eppendorf tube without disturbing the pellet. The tissue weight was recorded for the normalization analysis of cytokines.

## Luminex cytokine assay

Tissue lysate was used for analysis of the cytokine profile (SDF-1$\alpha$, IL-2, IL-4, IP-10, IL-10, IL-17A, IFN-$\gamma$, TNF-$\alpha$, I-TAC, granzyme B, sCD40L) using a Luminex assay. The Singleplex Luminex Protein Assay Kit for each cytokine was from ThermoFisher Scientific. Antibody-specific capture magnetic beads were added to wells of a 96-well plate. Samples and protein standards were then placed into the microplate wells. After incubation, the beads were washed using a handheld magnet and were resuspended in secondary detection antibody solution followed by washing and addition of streptavidin-RPE. The beads were then washed again for analysis on a FlexMAP 3D (ThermoFisher Scientific).

## EDTA treatment of HT-29 cells

HT-29 cells were collected from T75 flasks (Thermo Scientific) using different concentrations of EDTA (0.1; 0.2; 0.5; 1 and 2 mM) for 10 min at 37°C. Cells were blocked with 10% BSA/10% normal mouse serum (NMS)/10% FCS for 15 min at 4°C. Cells were then incubated with biotin-labeled NC410 (10 µg/mL; NextCure) in PBS + 1% BSA buffer for 30 min RT. After washing with PBS + 1% BSA, by spinning down for 5 min at 1500 rpm at RT, cells were incubated with streptavidin APC (eBioscience) diluted in PBS + 1% BSA buffer for 20 min at 4°C. Cells were washed again and measured on a FACSCanto (BD Biosciences).

## Collagenase treatment of HT-29 cells

HT-29 cells were collected from T75 flasks (Thermo Scientific) using 0.1 mM EDTA for 10 min at 37°C. Cells were washed and treated with 40 U Collagenase (from *Clostridium histolyticum*; Sigma) for different time points (5, 10, 15, 20 and 30 min) at 37°C. Cells were washed by spinning down for 5 min at 1500 rpm at RT and blocked with 10% BSA/10% NMS/10% FCS for 15 min at 4°C. Cells were washed and incubated with biotin-labeled NC410 (10 µg/mL; NextCure) in PBS + 1% BSA buffer for

30 min RT. After washing with PBS + 1% BSA, by spinning down for 5 min at 1500 rpm at RT, cells were incubated with streptavidin APC (eBioscience) diluted in PBS + 1% BSA buffer for 20 min at 4°C. Cells were washed again and measured on a FACSCanto (BD Biosciences).

## ADCC assay

ADCC with $^{51}$Cr-labeled target cells was described previously. Briefly, HT-29 target cells were labeled with 100 µCi (3.7 MBq) $^{51}$Cr for 3 hr in complete medium. After extensive washing, cells were adjusted to $10^5$/mL. HT-29 cells were then incubated with NC410 or isotype control for 30 min. Different effector to target ratios (E:T) were made by adding increasing amounts of PBMCs to NC410 or isotype-treated HT-29 cells per well (a fixed amount of 10.000 tumor cells was used) in round-bottom microtiter plates (Corning). After 24 hr of incubation at 37°C, $^{51}$Cr release was measured in counts per minute (cpm). The percentage of specific lysis was calculated using the following formula: % lysis = [(counts of sample–minimum release)/(maximum release–minimum release)] × 100. Target cells with PBMCs in complete medium or supplemented with 5% Triton X-100 (Roche Diagnostics) were used to determine minimum and maximum release, respectively.

## Immunofluorescence staining

5000 HT-29 cells were seeded in black Falcon clear flat-bottom 96-well plates and cultured for 3 days. Cells were fixed with 4% paraformaldehyde for 15 min at RT and blocked with 5% BSA in PBS for 1 hr at RT. Cells were then incubated with isotype control (NextCure), biotin-labeled NC410 (10 µg/mL, NextCure) or pan-collagen antibody (ThermoFisher Scientific) diluted in PBS + 1% BSA buffer for 1 hr at RT. After thoroughly washing with PBS, the slides were incubated with anti-human IgG1-AlexaFluor 594 or streptavidin AlexaFluor 594 (ThermoFisher Scientific) diluted in PBS + 1% BSA buffer for 30 min at RT. Slides were finally washed and mounted with DAPI VectaShield hardset (Vector Lab) and allowed to settle before image acquisition on a Zeiss fluorescence microscopy (Zeiss) using Zen software and ImageJ.

## Collagen degradation peptide analysis

The mouse serum samples were assessed for collagen-degraded peptides (PRO-C3, C3M, C4M, C4G, PRO-C6, C6M, VICM, reC1M, PRO-C23) by immunoassays by Nordic Bioscience as previously described (*Jensen et al., 2018*; *Rønnow et al., 2020*).

## Tumor specimens for immunohistochemistry

Specimens of seven selected tumor types were included for analysis: HNSC, GBM, SKCM, NSCLC, HGSC, PDAC and STAD. Of each tumor type, in agreement with the ethical committee of the UMCU, formalin-fixed paraffin-embedded (FFPE) material of 9–10 tumor specimens and 5 healthy specimens was collected from the tissue biobank (research protocol 17-786). The cohort of HNSC tumors comprised tongue tumors with a diameter of 10 mm or larger. Only melanomas with a Breslow thickness of 0.5 mm or higher were included. Tissue was obtained from primary tumors. Patients did not receive any systemic treatment or radiotherapy before the tumor specimens were obtained. Healthy specimens were preferably separate tissue blocks obtained from the same patients the tumor specimens were obtained from. In case of glioblastoma, healthy specimens were collected postmortem.

## Immunohistochemistry

The following stainings were performed on consecutive slides of all tumor specimens: H&E, Masson's Trichrome, LAIR-1, NC410, CD45, CD3, CD68 and CD163. The LAIR-1, CD45, CD3, CD68 and CD163 stainings were performed using a Ventana Bench Mark XT Autostainer (Ventana Medical Systems, Tucson, AZ, USA). Stainings were performed on 4 µm sections of each tissue block. NC410 was biotinylated and used for manual, IHC staining. Antigen retrieval was performed by incubating the slides at 100°C for 24 or 64 min in either EDTA or Citrate buffer, as indicated in the table below. For all assays, the sections were incubated with the primary antibody for 1 hr at RT. Subsequently, the sections were incubated with HRP-labeled secondary antibody, developed using $H_2O_2$ and DAB and counterstained with hematoxylin.

| Antibody | Supplier | Clone | Lot no | Pretreatment | Dilution |
|---|---|---|---|---|---|
| LAIR-1 (polyclonal) | ATLAS antibodies | - | A103834 | 24' Citrate pH 6.0 | 1:500 |
| LAIR-1 (monoclonal) | BD Pharmingen | DX26 | 550810 | 64' EDTA pH 9.0 | 1:100 |
| NC410 | NextCure | - | - | 24' Citrate pH 6.0 | 1:500 |
| CD45 | DAKO | PD7/26 + 2B11 | 110 | 24' EDTA pH 9.0 | 1:25 |
| CD3 | DAKO | - | 20042625 | 24' EDTA pH 9.0 | 1:100 |
| CD68 | Novocastra | KP1 | 6009203 | 24' EDTA pH 9.0 | 1:1600 |
| CD163 | Novocastra | 10D6 | 6064616 | 24' EDTA pH 9.0 | 1:800 |

## Immunohistochemistry analysis

All slides were digitalized using the Aperio Scanscope XT slide scanner. Evaluation of stained tissue slides was performed using QuPath (version 0.2.0) software. The percentage of tumor tissue binding NC410 was calculated by annotation of the tumor in the tissue slide and quantifying the percentage of DAB-stained tissue by using the Pixel Classifier.

Scoring of the immune phenotype was based on the presence and distribution of CD3-positive lymphocytes (*Chen and Mellman, 2017*). Tumors with CD3$^+$ cells in the tumor fields were scored 'inflamed'; tumors with CD3$^+$ cells in their stroma but without or with a relatively low amount of CD3$^+$ in the tumor fields were scored 'immune-excluded'; tumors with a lack of CD3$^+$ cells in their stroma as well as in their tumor fields were scored 'immune-desert'.

For quantification of the immune cell counts, regions of interest (ROIs) were annotated on the tissue slides by drawing circles with a diameter of 600 µm at five random spots within the NC410 binding part of the tumor. Positive cells were quantified using the Positive Cell Detection tool. For each staining, stain vectors and DAB cutoffs were determined based on a representative slide; the settings were kept the same for all slides. Tumor area within the circles that was lost during staining procedure or that was negative for NC410 binding was excluded from the analysis.

The percentage of NC410 binding to tissue within a tumor was calculated by dividing the stained area by the total tumor area. The immune cell counts for each tumor were calculated by dividing the total number of positive cells within the five ROIs by the total surface in mm$^2$ of these ROIs.

## Statistics

Data were graphed and analyzed using GraphPad Prism 8.0 (GraphPad software). Data are presented as the mean ± standard deviation (SD). The significance was analyzed with unpaired, two-tailed Student's t-tests and two-way ANOVA followed by multiple comparison.

## Acknowledgements

We thank Michiel van der Vlist for his critical comments on the manuscript.

## Additional information

### Competing interests

Linjie Tian: LT, CS, AP, JS, JB, ZC, LL, SL and DF are employees from Nextcure. Nextcure holds a patent on NC410. (PCT/US20 17/0453 10). The other authors declare that no competing interests exist.

### Funding

| Funder | Grant reference number | Author |
|---|---|---|
| Dutch Research Council (NWO) | Vici 918.15.608 | Linde Meyaard |

The funders had no role in study design, data collection and interpretation, or the decision to submit the work for publication.

### Author contributions
M Ines Pascoal Ramos, Linjie Tian, Formal analysis, Investigation, Methodology, Writing - original draft, Writing - review and editing; Emma J de Ruiter, Chang Song, Ana Paucarmayta, Akashdip Singh, Saskia V Vijver, Formal analysis, Investigation, Writing - review and editing; Eline Elshof, Formal analysis, Methodology, Writing - review and editing; Jahangheer Shaik, Data curation, Formal analysis, Investigation, Writing - review and editing; Jason Bosiacki, Christina Jensen, Investigation, Writing - review and editing; Zachary Cusumano, Conceptualization, Project administration, Writing - review and editing; Nicholas Willumsen, Morten A Karsdal, Formal analysis, Writing - review and editing; Linda Liu, Sol Langermann, Stefan Willems, Conceptualization, Formal analysis, Writing - review and editing; Dallas Flies, Linde Meyaard, Conceptualization, Formal analysis, Supervision, Funding acquisition, Writing - review and editing

### Author ORCIDs
M Ines Pascoal Ramos ⬚ https://orcid.org/0000-0003-3644-6517
Akashdip Singh ⬚ http://orcid.org/0000-0001-5326-8826
Dallas Flies ⬚ https://orcid.org/0000-0002-9280-2080
Linde Meyaard ⬚ https://orcid.org/0000-0003-0707-4793

### Ethics
Human subjects: Peripheral Blood Mononuclear Cells (PBMCs) were isolated from blood of healthy donors in agreement with ethical committee of the University Medical Center Utrecht (UMCU) and after written informed consent from the subjects in accordance with the Declaration of Helsinki. Specimens of seven selected tumor types were included for analysis: head and neck squamous cell carcinoma (HNSC), glioblastoma (GBM), melanoma, non-small-cell lung carcinoma (NSCLC), high-grade serous carcinoma (HGSC), pancreatic ductal adenocarcinoma (PDAC), and stomach adenocarcinoma (STAD). Of each tumor type, in agreement with the ethical committee of the UMCU, formalin fixed, paraffin embedded (FFPE) material of 9-10 tumor specimens and five healthy specimens was collected from the tissue biobank (research protocol 17-786).
Animal experimentation: All mouse studies were performed at NextCure based on Institutional Animal Care and Use Committee standards according to the protocols of NextCure Animal (NCA) Study 164 (NCA#164 for Figure 3), NCA#122 (for Figure 4), NCA#209 (for Figure 5), NCA#217 (for Figure 6) and NCA#270 (for Figure supplement 3).

### Decision letter and Author response
Decision letter https://doi.org/10.7554/eLife.62927.sa1
Author response https://doi.org/10.7554/eLife.62927.sa2

## Additional files
### Supplementary files
• Source code 1. Survival curves comparing individuals with high collagen expression against individuals with low collagen expression. Expression values corresponding to 43 collagen genes in tumor were obtained and averaged for each individual. The collagen expression from various individuals was divided into four quantiles. The individuals with expression values less than or equal to the first quantile were regarded as low expression, and those with values greater than or equal to the third quantile were considered as high expression. The association between expression (high/low) and their survival was assessed using Kaplan–Meier method. The survival curves were drawn using ggsurvplot function in the survminer package in R. The cancers where p-value was less than 0.05 were considered significant and are displayed.

• Source code 2. Average collagen expression in normal and tumor across various cancers. The expression values corresponding to 43 different collagen proteins are queried from the TCGA database. TCGA consists of 33 projects corresponding to 33 different cancers, which can individually be queried for expression of genes. For each cancer, expression of collagen genes in cancer and normal (where available) was queried individually. The expression across all collagens was averaged in tumor and normal and displayed as a bar graph in different cancers.

• Source code 3. Average LAIR-1/2 expression in normal and tumor across various cancers. Expression of LAIR-1/2 is queried from the TCGA database. Expression of LAIR-1/2 in normal and tumor is displayed as bar graphs in each cancer.

• Source code 4. Survival curves comparing individuals with high LAIR-1/2 expression against individuals with low LAIR-1/2 expression. LAIR-1/2 expression was obtained for each individual and divided into four quantiles. The individuals with expression values less than or equal to the first quantile were regarded as low expression, and those with values greater than or equal to the third quantile were considered as high expression. The association between expression (high/low) and their survival was assessed using Kaplan–Meier method, and the cancers where p-value was less than 0.05 are displayed.

• Source code 5. Survival curves comparing individuals with high collagen, high LAIR-1 expression against individuals with low collagen, low LAIR-1 expression. Two-way analysis involved dividing average collagen expression across individuals into low and high categories based on the quantiles as *Source code 4*. Similarly, LAIR-1 expression was also divided into low and high categories based on quantiles. The individuals with high LAIR-1 and high collagen expression were considered high group, and those with low LAIR-1 and low collagen expression were considered low group. Survival based on expression of these two categories was evaluated using Kaplan–Meier method, and the cancers where p-value was less than 0.05 is displayed.

• Transparent reporting form

## Data availability
Source codes were provided for Figure 1, Figure 2 and Figure supplement 1.

The following previously published dataset was used:

| Author(s) | Year | Dataset title | Dataset URL | Database and Identifier |
|---|---|---|---|---|
| Xu X, Zhang Y, Williams J, Antoniou E, McCombie WR, Wu S, Zhu W, Davidson NO, Denoya P, Li E | 2013 | Parallel comparison of Illumina RNA-Seq and Affymetrix microarray platforms on transcriptomic profiles generated from 5-aza-deoxy-cytidine treated HT-29 colon cancer cells and simulated datasets | https://www.ncbi.nlm.nih.gov/geo/query/acc.cgi?acc=GSE41586 | NCBI Gene Expression Omnibus, GSE41586 |

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

## Appendix 1

### Bioinformatics (extended)

For collagen gene expression analysis of HT-29 cells, data was acquired from the public dataset GSE41586 (https://pubmed.ncbi.nlm.nih.gov/23902433/). Raw count data of untreated HT-29 cells was retrieved and normalized using the DESeq2 package (v1.28.1) in R (v4.0.2). Data was then log2 transformed and plotted using the ggplot package (v3.3.2).

### Immunofluorescence staining

5000 HT-29 cells were seeded in black Falcon clear flat-bottom 96-well plates and cultured for 3 days. Cells were fixed with 4% paraformaldehyde for 15 min at RT and blocked with 5% BSA in PBS for 1 hr at RT. Cells were then incubated with isotype control (NextCure), biotin-labeled NC410 (10 µg/mL, NextCure) or pan-collagen antibody (ThermoFisher) diluted in PBS + 1% BSA buffer for 1 hr at RT. After thoroughly washing with PBS, the slides were incubated with anti-human IgG1-Alexa-Fluor 594 or streptavidin AlexaFluor 594 (Life Technologies–ThermoFisher Scientific) diluted in PBS + 1% BSA buffer for 30 min at RT. Slides were finally washed and mounted with DAPI Vecta-Shield hardset (Vector Lab) and allowed to settle before image acquisition on a Zeiss fluorescence microscopy (Zeiss) using the Axiovision software (Zeiss).

### EDTA treatment of HT-29 cells

HT-29 cells were collected from T75 flasks (Thermo Scientific) using different concentrations of EDTA (0.1; 0.2; 0.5; 1 and 2 mM) for 10 min at 37C. Cells were blocked with 10% BSA/10% NMS/10% FCS for 15 min at 4°C. Cells were then incubated with biotin-labeled NC410 (10 µg/mL; Next-Cure) in PBS + 1% BSA buffer for 30 min RT. After washing with PBS + 1% BSA, by spinning down for 5 min at 1500 rpm at RT, cells were incubated with streptavidin APC (eBioscience) diluted in PBS + 1% BSA buffer for 20 min at 4°C. Cells were washed again and measured on a FACSCanto (BD Biosciences).

### Collagenase treatment of HT-29 cells

HT-29 cells were collected from T75 flasks (Thermo Scientific) using 0.1 mM EDTA for 10 min at 37°C. Cells were washed and treated with 40 U Collagenase (from *Clostridium histolyticum*; Sigma) for different time points (5, 10, 15, 20 and 30 min) at 37°C. Cells were washed by spinning down for 5 min at 1500 rpm at RT and blocked with 10% BSA/10% NMS/10% FCS for 15 min at 4°C. Cells were washed and incubated with biotin-labeled NC410 (10 µg/mL; NextCure) in PBS + 1% BSA buffer for 30 min RT. After washing with PBS + 1% BSA, by spinning down for 5 min at 1500 rpm at RT, cells were incubated with streptavidin APC (eBioscience) diluted in PBS + 1% BSA buffer for 20 min at 4°C. Cells were washed again and measured on a FACSCanto (BD Biosciences).

