## [Decision Letter]

**Acceptance summary:**

Collagen is a major component of extracellular matrix. The authors have identified a high-affinity inhibitory collagen receptor LAIR-1 and a soluble decoy receptor LAIR-2 (with even higher binding affinity to collagen), which can be therapeutically targeted to block tumor progression. Dr Meyaard and colleagues have also generated a dimeric LAIR-2 human IgG1 Fc fusion protein NC410 for therapeutic use. With humanized mouse models engrafted with functional human immune systems (PBMC), they have explored the anti-cancer efficacy of NC410 and revealed its impact on modulating immune responses. Furthermore, they extended this study to identify biomarkers of predictive value for NC410-based anti-cancer therapy.

**Decision letter after peer review:**

Thank you for submitting your article "Cancer immunotherapy by NC410, a LAIR-2 Fc protein blocking LAIR-collagen interaction" for consideration by *eLife*. Your article has been reviewed by 3 peer reviewers, one of whom is a member of our Board of Reviewing Editors, and the evaluation has been overseen by Carla Rothlin as the Senior Editor. The reviewers have opted to remain anonymous.

The reviewers have discussed the reviews with one another and the Reviewing Editor has drafted this decision to help you prepare a revised submission.

Summary:

Collagen is a major component of extracellular matrix. The authors have identified a high-affinity inhibitory collagen receptor LAIR-1 and a soluble decoy receptor LAIR-2 (with even higher binding affinity to collagen), which can be therapeutically targeted to block tumor progression. They have generated a dimeric LAIR-2 human IgG1 Fc fusion protein NC410 for therapeutic use. With humanized mouse models engrafted with functional human immune systems (PBMC), they have explored the anti-cancer efficacy of NC410 and revealed its impact on modulating immune responses. Furthermore, they extended this study to identify biomarkers of predictive value for NC410-based anti-cancer therapy. This study is interesting, but mechanistic exploration and in-depth immune characterization should be largely improved to support their conclusions.

Essential revisions:

1. Figure 1 is quite confusing for readers. These correlative analyses don't seem to support the major conclusions of this study. Although the expression level of collagen, LAIR1 and LAIR2 in multiple cancer tissues significantly differ from that in matched normal tissues (Figure 1A), these parameters didn't exhibit consistent prognostic values for the same type of cancer (Figure 1B). High expression of LAIR-1 is associated with lower survival, but this only holds true in 4 of 21 cancers. Conversely, high LAIR-2 mRNA expression was correlated with increased survival in 6 of 30 cancers. There was no overlap in these cancer types. It would be much more helpful if the authors can show that in the same cohorts of cancer patients, high expression of Collagen, high expression of LAIR1, and/or low expression of LAIR2 correlate with poor outcomes. If analyses of this potential biomarker panel in Figure 1 can be linked with the pathologic analyses performed in Figure 6, it will largely strengthen the importance of this study.

2. Similarly, for the TCGA analyses in Figure 1, it will be very helpful to explore the correlations of the collagens, LAIR-1 and LAIR-2 with immune phenotypes as defined by mRNA signatures. This would better define if the collagen/LAIR-1 pathway is more important in inflamed, but immunosuppressed tumors, or in cold tumors.

3. NC410 exhibited anti-tumor effects in two humanized mouse model. A more robust immune profiling of the tumor microenvironment would be necessary, particularly for the myeloid compartment (such as MDSCs, tumor-associated macrophages and their effector molecules) and T cells in depth (memory, effector, exhaustion markers). It remains unknown whether elevated frequency of CD4^+^T and CD8^+^T cells, increased secretion of IFN-g and TNF-a by T cells, augmented expression of cytokines, chemokines did account for delayed tumor progression. Functional validation of these altered immune cell subsets and corresponding effector molecules will substantially improve the quality of this study.

4. The authors claimed that LAIR2 is responsible for collagen-induced immunosuppression. Which immunosuppressive cell populations can be induced/activated/expanded by collagen? Does NC410 reverse these changes?

5. There are multiple receptors for collagens. Is LAIR1 the dominant immunosuppressive receptor for collagen? Does NC410 specifically block LAIR1-mediated immunosuppression?

6. Does NC410 influence the synthesis of collagen? Is dimeric LAIR-2 more stable than the monomer form? What is the function of IgG1 Fc tail? It is possible that Fc containing NC410 cause the ADCC effect on collagen-producing cells (including tumor cells) in vivo. HT29 cell and PBMC coculture experiments didn't rule out the possibility of NC410-induced ADCC in vivo.

7. Does repeated use of NC410 cause detrimental side-effects on healthy collagen-enriched tissues? Do the authors have any toxicity data on the in vivo use of this new Fc-fusion protein? It would be very helpful at their first description for potential clinical development.

[Editors' note: further revisions were suggested prior to acceptance, as described below.]

Thank you for resubmitting your work entitled "Cancer immunotherapy by NC410, a LAIR-2 Fc protein blocking human LAIR-collagen interaction" for further consideration by *eLife*. Your revised article has been evaluated by Carla Rothlin as the Senior Editor, and a Reviewing Editor.

The manuscript has been improved but there are some remaining issues that need to be addressed, as outlined below:

1. Gating strategies should be shown for the analyses of naive and memory T cell subsets in Figure 3.

2. Figure 6c is not necessary. The implication of these parameters can be described in the main text.

3. Due to low-quality images in Figure 7 and related supplemental data, it is difficult to judge the specificity of IHC staining. Please provide images with higher magnifications and resolution. Also, please avoid using the same image in different figures.

---

## [Author Response]

Essential revisions:1. Figure 1 is quite confusing for readers. These correlative analyses don't seem to support the major conclusions of this study. Although the expression level of collagen, LAIR1 and LAIR2 in multiple cancer tissues significantly differ from that in matched normal tissues (Figure 1A), these parameters didn't exhibit consistent prognostic values for the same type of cancer (Figure 1B). High expression of LAIR-1 is associated with lower survival, but this only holds true in 4 of 21 cancers. Conversely, high LAIR-2 mRNA expression was correlated with increased survival in 6 of 30 cancers. There was no overlap in these cancer types. It would be much more helpful if the authors can show that in the same cohorts of cancer patients, high expression of Collagen, high expression of LAIR1, and/or low expression of LAIR2 correlate with poor outcomes. If analyses of this potential biomarker panel in Figure 1 can be linked with the pathologic analyses performed in Figure 6, it will largely strengthen the importance of this study.

We have restructured figure 1 for clarity (see new figure 1 and Figure 1-Sup. Figure 1). To support targeting the collagen:LAIR-1 pathway in cancer, we looked at the overall survival of total collagens (Figure 1A) and LAIR-1 individually (Figure 1B). We also analyzed total collagens combined with LAIR-1 for overall survival (Figure 1C). The expression of collagens, LAIR-1 and LAIR-2 in healthy and tumor tissue was moved to Figure 1-Sup figure 1. Overall survival analysis of LAIR-2 high expression associated with improved overall survival was moved to Figure 2 for organizational clarity.

These analyses have the restriction that it is limited to mRNA expression and not protein. However, we utilized these data to support targeting the LAIR-collagen axis for cancer. Whereas most other studies evaluate specific collagens, we uniquely evaluated all 43 collagen chains together for overall survival to assess the “net effect” of collagens in relation to overall survival.

The overall survival analysis based on LAIR-1 RNA expression does not overlap with total collagens, other than low grade glioma, but led to a worse prognosis in 4 of 21 cancers. This may suggest additional factors influence outcome. Therefore, we combined total collagens and LAIR-1 for analysis of overall survival. We found that 10 of 21 cancers had decreased survival when both collagen and LAIR-1 were highly expressed (upper quantile). Interestingly, cancer types in the LAIR-1 only analysis, but not collagen only analysis, overlapped with the combined collagen+ LAIR-1 analysis. A few new tumor types also emerged that were not present when evaluated individually. Again, this analysis was performed in order to support targeting the LAIR-1 pathway in cancer, not necessarily to select specific cancer indications for clinical trials.

Next, our hypothesis is that LAIR-2, a natural soluble decoy capable of blocking membrane LAIR-1 mediated immune inhibition could be used as a “supplemental” checkpoint blockade therapy. Therefore, we analysed if increased LAIR-2 was associated with improved survival in any cancer. Despite the generally low expression of LAIR-2 in comparison to LAIR-1 in cancer, we did observe improved survival in 6 of 21 cancer types. Once again there was relatively little overlap with total collagen. However, this finding did support our hypothesis.

Together, while overall survival analyses based on RNA expression did not overlap to a large extent, the purpose was to support targetability of the collagen:LAIR pathway, and potentially serve as a guide in selection of cancer types for further evaluation by IHC. To this end, emphasis was placed on cancer types in which high LAIR-2 expression associated with improved survival (HNSCC, STAD and SKCM). In addition, we selected LUSC, PDAC and HGSC to be evaluated by IHC. In figure 6, we show that LAIR-1^+^ cells are located in collagen-rich areas where NC410 can bind suggesting that the location of these molecules and cells, rather than overall RNA levels, may be more important for patient selection.

Text was added to the manuscript to address this point.

2. Similarly, for the TCGA analyses in Figure 1, it will be very helpful to explore the correlations of the collagens, LAIR-1 and LAIR-2 with immune phenotypes as defined by mRNA signatures. This would better define if the collagen/LAIR-1 pathway is more important in inflamed, but immunosuppressed tumors, or in cold tumors.

As discussed in the question 1 response, we analyzed total collagens combined with LAIR-1 for overall survival to determine if there is overlap or unique indications in comparison to collagens or LAIR-1 alone. We found that high expression of total collagens with LAIR-1 resulted in poor reduced overall survival that overlapped with LAIR-1 analysis, but also that additional indications were identified by collagens alone or LAIR-1 alone.

The reviewers’ question on immune phenotypes we address in Figure 6, where we concluded that patients presenting immune excluded tumors would benefit the most from LAIR targeting since LAIR-1^+^ cells seem to be enriched in collagen dense areas surrounding the tumor islets where NC410 binds. By analysing mRNA signatures rather that spatial context we would not be able to show this.

Text was added to the Discussion section to address this point.

3. NC410 exhibited anti-tumor effects in two humanized mouse model. A more robust immune profiling of the tumor microenvironment would be necessary, particularly for the myeloid compartment (such as MDSCs, tumor-associated macrophages and their effector molecules) and T cells in depth (memory, effector, exhaustion markers). It remains unknown whether elevated frequency of CD4^+^T and CD8^+^T cells, increased secretion of IFN-g and TNF-a by T cells, augmented expression of cytokines, chemokines did account for delayed tumor progression. Functional validation of these altered immune cell subsets and corresponding effector molecules will substantially improve the quality of this study.

We have performed extra studies and added additional data to address these points.

This manuscript focuses on the role of NC410 in the context of T cells. Therefore, peripheral blood mononuclear cell (PBMC) humanized mice (Hu-PBMC-NSG) provide a useful platform for investigating the modulation of the human T cell responses and antitumor responses. As previously reported (Ali et al., PLoS ONE, 2012), human T cells but not myeloid cells expand in this model and almost all the human CD45^+^ cells engrafted are T cells. Nevertheless, the role of NC410 on myeloid and other TME associated immune cell subsets is of great interest to us and will be the focus of a follow-up study.

We revised figure 3 to address the role of NC410 on CD8^+^ and CD4^+^ T cells in both non-tumor (xeno-GVHD) and tumor models in hu-PBMC-NSG mice. In figure 3A and B, we show that most T cells (80-100%) in the hu-PBMC-NSG model acquire effector memory phenotype in the spleen as early as 6 days after PBMC injection in NSG mice. This is similar to previous reports (Ali et al., PLoS ONE, 2012). The percentage of effector memory CD4^+^ or CD8^+^ T cells does not change between doses of NC410 vs vehicle control. However, as shown in Figure 3C and E, NC410 dose-dependently increase the absolute cell counts of CD4^+^ Effector memory (EM) and CD8^+^ effector T cells in the spleen on day 6. Unfortunately, we did not assess cytokine levels in this experiment, but this was evaluated in Figure 5.

We also added additional data in a new Figure 4 for the P815 tumor model in which the mouse tumor cell line P815 (xeno tumor) is subcutaneously injected into the flank of NSG mice with adoptive transfer of human PBMCs. A titration of NC410 shows that NC410 promotes the expansion of both CD4^+^ and CD8^+^ T cells, but that the expansion follows different kinetics dependent on the dose: 10 mg/kg induces more rapid expansion, while 1 mg/kg induces sustained expansion. Interestingly, the anti-tumor activity observed in this model corresponds to the 1 mg/kg dose in which sustained T cell expansion is observed. This supports the role of T cells in mediating anti-tumor immunity.

Importantly, to address the requirement of T cells for the anti-tumor effect observed in the presence of NC410, we performed an additional HT-29 tumor model in NSG mice similar to the experiments shown in Figure 5 (Figure 3-Sup. Figure 3). However, in this experiment, we adoptively transferred total PBMCs or T cell depleted PBMCs. In this experiment we show that NC410 required T cells for anti-tumor activity, as shown by tumor growth measurements and pictures of tumors at endpoint (day 53). This experiment demonstrates that NC410 anti-tumor immunity in the HT-29 tumor model is mediated through T cell enhancement.

4. The authors claimed that LAIR2 is responsible for collagen-induced immunosuppression. Which immunosuppressive cell populations can be induced/activated/expanded by collagen? Does NC410 reverse these changes?

The authors believe there may be a typo on this question. LAIR-2 is not responsible for collagen-induced immunosuppression. Rather, LAIR-2 is the basis of our immunotherapy strategy to block and reverse LAIR-1 mediated immunosuppression by tumor collagens.

We propose that LAIR-1 induced immune suppression is due to decrease of T cell activity and/or migration. Collagen has been previously shown to have a direct impact on T cell activation via LAIR-1 by inhibiting T-cell receptor (TCR) signaling (Meyers L, et al; JBC, 2020) and collagen has been shown to inhibit T cell migration (Kuczek et al., J Immunother Cancer, 2019), however this has not been specifically attributed to LAIR-1 signaling. Our in vitro studies (Figure 2) demonstrate that collagen binding to LAIR-1 induced LAIR-1 signaling in a reporter cell line can be blocked by NC410. Our in vivo studies demonstrate increased T cell expansion and effector activity in the presence of NC410. However, at this time we cannot prove that the NC410 mediated effect is dependent solely on blockade of LAIR-1:Collagen interactions. Interestingly, a similar molecule to NC410 (Xu L, et al; OncoImmunology, 2020) has been shown to revert collagen induced LAIR-1 suppression of IFN-γ and TNF-α cytokine production in an in vitro system.

5. There are multiple receptors for collagens. Is LAIR1 the dominant immunosuppressive receptor for collagen? Does NC410 specifically block LAIR1-mediated immunosuppression?

Diverse families of receptors, including integrins, receptor tyrosine kinases, and immunoglobulin-like receptors have been shown to bind collagens as their cognate ligands. We cannot exclude that NC410 blocks other receptors or has additional functions besides outcompeting LAIR-1. In fact, our group has previously shown that LAIR-2 Fc can efficiently inhibit complement activation via classical and lectin pathways (Olde Nordkamp, et al; J Innate Immun. 2014). Therefore, it is likely that NC410 has additional mechanisms of action that may be LAIR-1 independent. This is a major topic of ongoing research for future studies due to the importance of mechanistic understanding and biomarker identification as part of NextCure’s ongoing clinical trial.

We addressed this point in the discussion of the manuscript.

6. Does NC410 influence the synthesis of collagen? Is dimeric LAIR-2 more stable than the monomer form? What is the function of IgG1 Fc tail? It is possible that Fc containing NC410 cause the ADCC effect on collagen-producing cells (including tumor cells) in vivo. HT29 cell and PBMC coculture experiments didn't rule out the possibility of NC410-induced ADCC in vivo.

Our current study results have identified an effect of NC410 leading to changes in specific collagen degradative products, which may be an important pharmacodynamic (PD) biomarker for clinical trials. We are currently studying the biological relevance of NC410 on collagen production, degradation and organization with collaborators. This data will be part of a follow up study.

NC410 is very stable based on production and manufacturing of material for the ongoing clinical trial with NC410 (Phase 1 clinical trial was initiated with NC410 on June 10, 2020; ClinicalTrials.gov Identifier: NCT04408599). Pharmacokinetics will be assessed as part of Ph1 trial. Stability of NC410 has not been compared to monomeric LAIR-2. We previously reported that dimeric LAIR-2 is a better competitor for LAIR-1-collagen interaction than monomeric LAIR-2 (Olde Nordkamp et al., Arthritis Rheum, 2011)

We agree with the reviewer that HT-29 cell and PBMC co-culture experiments do not completely rule out the possibility of NC410-induced ADCC in vivo. However, several additional lines of evidence support the function of NC410 to be independent of ADCC, but dependent on an active, functional Fc (human IgG1).

First, P815 tumor cells for not express transmembrane collagen and thus in the P815 tumor model (Figure 4), NC410 does not bind to this tumor by flow cytometry analysis (Figure 4A and B), and therefore ADCC is unlikely to be a mechanism of action.

Secondly, in Figure 3-Sup figure 3, we tested NC410 in the absence of T cells and found that T cells are required for anti-tumor effect mediated by NC410, as discussed above, which indicated that ADCC is not a primary mechanism of action in vivo.

Therefore, we conclude that ADCC is not a primary mechanism of action of NC410, although we cannot rule out that ADCC may play a role in in vivo tumor killing for collagen expressing tumors.

We added this discussion to the revised manuscript.

To further address the question of what the Fc portion of NC410 does, we show in a new Figure 3- Sup. Figure 3 that LAIR-2 IgG1 protein with a “silenced” Fc (FES mutation) (Oganesyan et al., Biol Crystallogr. 2008) loses anti-tumor activity. This is in apparent contrast with a recently published study (Xu et al., Oncoimmunology, 2020) in which an engineered LAIR-2 Fc protein with an N297A to prevent binding to Fc receptors does have anti-tumor efficacy, However, this protein was further mutated to include T250Q/M428L mutations to increase binding to neonatal FcR (FcRn), which enhances in vivo availability of the protein. We cannot exclude that the lack of efficiency of the LAIR-2-FES we used is due to decreased in vivo availability of the protein.

7. Does repeated use of NC410 cause detrimental side-effects on healthy collagen-enriched tissues? Do the authors have any toxicity data on the in vivo use of this new Fc-fusion protein? It would be very helpful at their first description for potential clinical development.

No detrimental effects have been observed in mice, rats or cynomolgus macaques. Safety/toxicity studies in cynomolgus macaques were completed for IND filing and was found to be safe for a First-In-Human Ph1 clinical trial for NC410, which is currently ongoing. Safety and tolerability will continue to be monitored during dose escalation of NC410.

[Editors' note: further revisions were suggested prior to acceptance, as described below.]

The manuscript has been improved but there are some remaining issues that need to be addressed, as outlined below:1. Gating strategies should be shown for the analyses of naive and memory T cell subsets in Figure 3.

A new Figure 3 supplemental figure 1 was added where the gating strategy is shown.

2. Figure 6c is not necessary. The implication of these parameters can be described in the main text.

Figure 6c was removed and information regarding collagen degradation products was added to the respective figure legend.

3. Due to low-quality images in Figure 7 and related supplemental data, it is difficult to judge the specificity of IHC staining. Please provide images with higher magnifications and resolution. Also, please avoid using the same image in different figures.

Images in Figure 7 and related supplemental data were replaced by higher resolution images (good resolution up to 800% of magnification) and duplicate images were replaced.